# A FRET biosensor for necroptosis uncovers two different modes of the release of DAMPs

Shin Murai [1], Yoshifumi Yamaguchi [2], Yoshitaka Shirasaki [3,4], Mai Yamagishi [4], Ryodai Shindo [1], Joanne M. Hildebrand [5,6], Ryosuke Miura [1,7], Osamu Nakabayashi [1], Mamoru Totsuka [8], Taichiro Tomida [9], Satomi Adachi-Akahane [9], Sotaro Uemura [4], John Silke [5,6], Hideo Yagita [10], Masayuki Miura [11] & Hiroyasu Nakano [1,12]

Necroptosis is a regulated form of necrosis that depends on receptor-interacting protein kinase (RIPK)3 and mixed lineage kinase domain-like (MLKL). While danger-associated molecular pattern (DAMP)s are involved in various pathological conditions and released from dead cells, the underlying mechanisms are not fully understood. Here we develop a fluorescence resonance energy transfer (FRET) biosensor, termed SMART (a sensor for MLKL activation by RIPK3 based on FRET). SMART is composed of a fragment of MLKL and monitors necroptosis, but not apoptosis or necrosis. Mechanistically, SMART monitors plasma membrane translocation of oligomerized MLKL, which is induced by RIPK3 or mutational activation. SMART in combination with imaging of the release of nuclear DAMPs and Live-Cell Imaging for Secretion activity (LCI-S) reveals two different modes of the release of High Mobility Group Box 1 from necroptotic cells. Thus, SMART and LCI-S uncover novel regulation of the release of DAMPs during necroptosis.

[1] Department of Biochemistry, Toho University School of Medicine, 5-21-16 Omori-Nishi, Ota-ku, Tokyo 143-8540, Japan. [2] Hibernation Metabolism, Physiology, and Development Group, Environmental Biology Division, Institute of Low Temperature Science, Hokkaido University, Kita 19, Nishi 8, Kita-ku, Sapporo, Hokkaido 060-0819, Japan. [3] Precursory Research for Embryonic Science and Technology, Japan Science and Technology Agency, Chiyoda-ku, Tokyo 102-0075, Japan. [4] Department of Biological Sciences, Graduate School of Science, The University of Tokyo, 7-3-1 Bunkyo-ku, Tokyo 113-0033, Japan. [5] Division of Cell Signaling and Cell Death, The Walter and Eliza Hall Institute of Medical Research, Parkville, VIC 3052, Australia. [6] Department of Medical Biology, University of Melbourne, Parkville, VIC 3050, Australia. [7] Laboratory of Molecular Biology and Immunology, Department of Biological Science and Technology, Faculty of Industrial Science and Technology, Tokyo University of Science, 6-3-1 Niijuku, Katsushika-ku, Tokyo 125-8585, Japan. [8] Department of Food Science and Technology, Faculty of Applied Life Science, Nippon Veterinary and Life Science University, 1-7-1 Kyonancho, Musashino-shi, Tokyo 180-8602, Japan. [9] Department of Physiology, Toho University School of Medicine, 5-21-16 Omori-Nishi, Ota-ku, Tokyo 143-8540, Japan. [10] Department of Immunology, Juntendo University Graduate School of Medicine, 2-1-1 Hongo, Bunkyo-ku, Tokyo 113-8421, Japan. [11] Department of Genetics, Graduate School of Pharmaceutical Sciences, The University of Tokyo, 7-3-1 Bunkyo-ku, Tokyo 113-0033, Japan. [12] Host Defense Research Center, Toho University School of Medicine, 5-21-16 Omori-Nishi, Ota-ku, Tokyo 143-8540, Japan. Correspondence and requests for materials should be addressed to H.N. (email: hiroyasu.nakano@med.toho-u.ac.jp)

While apoptosis has been considered to be a typical form of programmed or regulated cell death, recent studies have revealed novel types of regulated or programmed cell death, including necroptosis, pyroptosis, and ferroptosis[1,2]. Necroptosis is morphologically similar to necrosis and can be induced by virus infection, or death ligands including tumor necrosis factor (TNF), Fas ligand, and TRAIL when caspase activation is blocked[3–5]. Upon stimulation with these death ligands, sequential phosphorylation and activation of receptor-interacting protein kinase (RIPK)1 and RIPK3 are induced. Activated RIPK3 phosphorylates mixed lineage kinase domain-like (MLKL), resulting in a conformational change of MLKL. Phosphorylated MLKL forms oligomers and translocates to biological membranes, resulting in the execution of necroptosis through pore formation and membrane rupture[6,7].

Danger-associated molecular pattern (DAMP)s are host-derived molecules released from dead cells through ruptured nuclear and cytoplasmic membranes. DAMPs include nuclear proteins such as High-Mobility Group Box (HMGB)1 and histones but also heat shock proteins, IL-1 family members, and ATP[8,9]. A previous study reported that HMGB1 is passively released from necrotic cells, but actively secreted from macrophages upon lipopolysaccharide (LPS)-stimulation[10,11]. Released HMGB1 mediates inflammation and also plays a crucial role in the development of septic shock through binding to Toll-like receptor (TLR)2 and 4, and the receptor for advanced glycan endproducts (RAGE)[10]. IL-33, a cytokine belonging to the IL-1 family, is released from the nucleus of necrotic epithelial cells and promotes T helper ($T_H$)2-type immune responses[12]. Intriguingly, IL-33 is cleaved by caspase 3 in apoptotic cells, resulting in the inactivation of its biological activity[12]. Moreover, we recently reported that large amounts of histone H3 are released from apoptotic hepatocytes and potentially involved in the development of endothelial cell injury in the livers and lungs[13]. Together, these results suggest that DAMPs are critically involved in inflammation, immune responses, and tissue injury. Thus, blockade of the release of DAMPs or neutralization of their biological activities might be novel strategies to treat various pathological conditions. However, it is not fully understood whether these DAMPs are actively secreted from dead cells or passively released from the nucleus or the cytosol to the extracellular space through the ruptured membrane. To investigate the mechanism(s) underlying the release of DAMPs from necroptotic cells, it is crucial to monitor the activation of MLKL and subsequent releases of DAMPs in a single cell. We previously developed a Live-Cell Imaging for Secretion activity (LCI-S) platform[14]. LCI-S enables us to immediately capture a cytokine secreted from a single cell, and a secreted cytokine is visualized by total internal reflection fluorescence microscopy-based fluorescence immunoassay (TIRFM-FIA).

Fluorescence resonance energy transfer (FRET) is a technique to measure the distance between two molecules based on energy transfer from a fluorescent donor dye to a fluorescent acceptor dye within a narrow distance[15]. FRET biosensors enable us to monitor dynamic changes of interaction, phosphorylation, and conformational changes of various signaling molecules in living cells. We previously developed a FRET biosensor termed SCAT (a Sensor for Caspase Activation based on FRET). SCAT1 and SCAT3 comprise peptides containing consensus caspase 1 and caspase 3 cleavage sequences between Venus and cyan fluorescent protein (CFP), enabling us to monitor pyroptosis and apoptosis, respectively[16–18]. Imaging of macrophages derived from mice stably expressing SCAT1 uncovered an intimate crosstalk between interleukin (IL)-1β release and membrane rupture along with caspase 1 activation[17]. On the other hand, analysis of SCAT3 transgenic mice revealed that the removal of apoptotic cells plays

a crucial role in neural tube closure during the development of murine embryos[16]. Together, these results indicate that live cell imaging of caspase activation in vitro and in vivo increases our understanding of pyroptosis and apoptosis under physiological and pathological conditions. In contrast to apoptosis or pyroptosis, it is unclear how signaling molecules crucial for necroptosis are regulated within a single cell during the execution of cell death.

To investigate the mechanisms underlying the release of DAMPs from necroptotic cells, we developed a FRET biosensor termed SMART (a Sensor for MLKL Activation by RIPK3 based on FRET). SMART monitored necroptosis, but not apoptosis or necrosis. An increase in the FRET/CFP ratio of SMART depended on endogenous RIPK3 and MLKL and correlated with phosphorylation of MLKL and RIPK3. More importantly, SMART monitored plasma membrane translocation of oligomerized MLKL. SMART in combination with imaging of the release of nuclear HMGB1 and LCI-S revealed that the release of HMGB1 was tightly correlated with the increase in the FRET/CFP ratio of SMART. We also found that there were two different modes of the release of HMGB1 from necroptotic cells. Thus, SMART and LCI-S uncover novel regulation of the release of DAMPs during necroptosis.

## Results

**Development of a FRET biosensor that monitors necroptosis.** To develop a FRET biosensor that monitors necroptosis, we focused on the conformational change of MLKL. Murine MLKL (mMLKL) is composed of an N-terminal four-helix bundle domain comprising 4 α helices and a C-terminal kinase-like (KL) domain. The KL domain of mMLKL is composed of 9 α helices and 7 β strands (Fig. 1a, Supplementary Fig. 1a). Crystal structures of the KL domain of MLKL show that it adopts an inactive conformation through a hydrogen bond between K219 and Q343 thereby preventing spontaneous oligomer formation[19]. RIPK3-dependent phosphorylation of serines and threonine in α4 helix promotes MLKL plasma membrane translocation and oligomerization, resulting in membrane permeabilization. Since the conformational change in the KL domain of mMLKL might increase the FRET efficiency, we designed intramolecular FRET biosensors comprising fragments of the KL domain between enhanced cyan fluorescent protein (ECFP) and modified yellow fluorescent protein (YPet) serving as FRET donor and acceptor, respectively. We inserted α1 to α9 (α19), α1 to α4 (α14), and α5 to α9 (α59) helices of the KL domain into a FRET backbone vector (Fig. 1a). A murine fibrosarcoma cell line, L929 cells undergo necroptosis when treated with TNF and a caspase inhibitor, z-VAD-fmk (TZ)[20]. To test our FRET biosensors, we transfected L929 cells with them and stimulated the cells with TZ. Transfection of α19 blocked TZ-induced necroptosis, and α19 showed a marginal increase in the FRET/CFP ratio in L929 cells (Supplementary Fig. 1b, c). In sharp contrast, α14, but not α59 showed a dramatic increase in the FRET/CFP ratio in cells undergoing necroptosis (Supplementary Fig. 1c). Maximum changes of the FRET/CFP ratio of α14 were significantly higher than those of α59 (Fig. 1b). Cotransfection experiments revealed that three biosensors interacted with RIPK3, and MLKL in the absence or presence of RIPK3 and without induction of necroptosis (Fig. 1c–e), suggesting that the interaction of FRET biosensors with RIPK3 or MLKL is not sufficient for FRET induction (Fig. 1d). Notably, Flag-RIPK3 was efficiently coimmunoprecipitated with Myc-MLKL in the presence of α19 (Fig. 1e), indicating that α19 did not block the binding of MLKL to RIPK3. Thus, we surmised that α19 might block the oligomerization of endogenous MLKL, thereby inhibiting TZ-induced necroptosis.

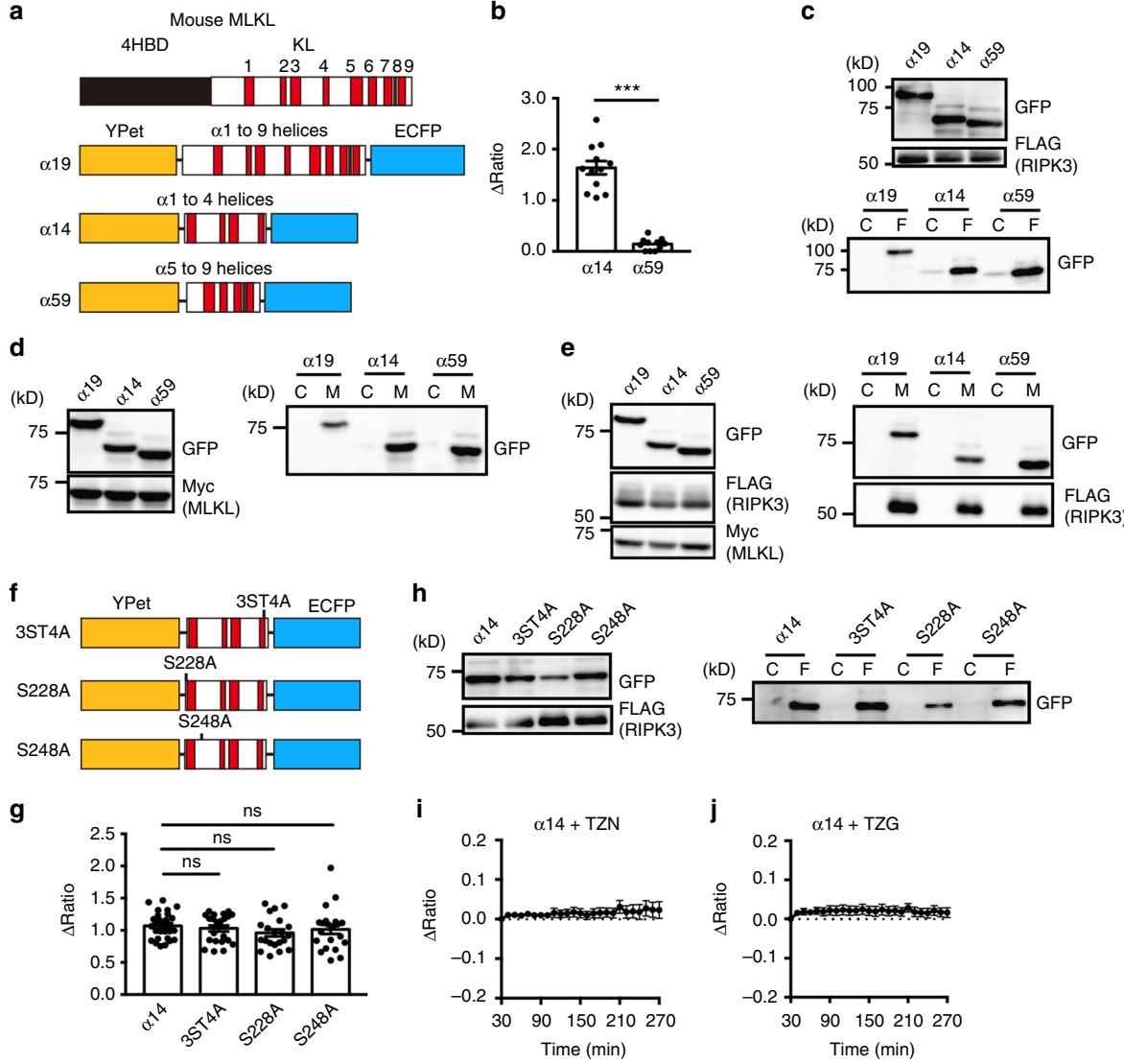

**Fig. 1** Development of a FRET biosensor that monitors necroptosis. **a** Domain structures of murine MLKL and designed FRET biosensors. 4HBD and KL indicate four-helical bundle and kinase-like domains, respectively. Red boxes indicate each α helix. **b** α14 monitors necroptosis. L929 cells were transiently transfected with α14 or α59, and then treated with TZ. The FRET/CFP ratio was calculated and analyzed as in Methods, and maximum changes of the FRET/CFP ratio are shown. Each dot indicates individual cell ($n = 12$ cells). Statistical significance was determined using the unpaired two-tailed Student's t test. ***$P < 0.001$. **c–e** HEK293T cells were transiently transfected with α19, α14, and α59 along with FLAG-RIPK3 (**c**), Myc-MLKL (**d**), or Myc-MLKL plus FLAG-RIPK3 (**e**). Otherwise indicated, murine RIPK3 and murine MLKL were used for experiments. At 24 h after transfection, cell lysates were immunoprecipitated with control, anti-FLAG (**c**) or anti-Myc antibodies (**d**, **e**), and then immunoprecipitates were analyzed by immunoblotting with the anti-GFP antibody. Expression of transfected constructs was verified by immunoblotting using total cell lysates. C, F, and M indicate control, anti-FLAG, and anti-Myc antibodies, respectively. **f** Diagrams of α14 mutants. 3ST4A indicates the quadruple mutant where S345, S347, T349, and S352 are replaced with alanines. **g** L929 cells were transiently transfected with the indicated mutants and then stimulated with TZ, and analyzed as in (**b**). Maximum changes of the FRET/CFP ratio. Each dot indicates individual cell ($n = 10$–14 cells). Statistical significance was determined using the one-way ANOVA test. ns, not significant. **h** HEK293T cells were transiently transfected with the indicated mutants along with FLAG-RIPK3. Cell lysates were immunoprecipitated and analyzed as in (**c**). **i**, **j** L929 cells were transiently transfected with α14 and then stimulated with TZ + Nec-I (TZN) (**i**) or TZ + GSK'872 (**j**), and the FRET/CFP ratio was analyzed as in (**b**). Kinetics of average of ΔFRET/CFP ratio is shown ($n = 9$ cells). Results are representative of two or three independent experiments, or pooled results of two independent experiments (**g**). Error bars indicate s.e.m.

Substitution of serines and threonine in the α4 helix of mMLKL (S345, S347, T349, and S352) with alanine (3ST4A for short) blocks TNF-induced necroptosis by preventing MLKL phosphorylation and oligomer formation[21,22]. Unexpectedly, TZ stimulation increased the FRET/CFP ratio in cells expressing α14 mutants 3ST4A, S228A, or S248A (Fig. 1f, g, Supplementary Fig. 1d), and these mutants bound to RIPK3 (Fig. 1h). Thus, while phosphorylation of MLKL is required for oligomerization of

MLKL, phosphorylation of α14 is not necessary for FRET induction. Nevertheless, a RIPK1 inhibitor, Necrostatin-1 (Nec-1) or a RIPK3 inhibitor, GSK'872 blocked necroptosis and abolished an increase in the FRET/CFP ratio in cells expressing α14 upon TZ stimulation (Fig. 1i, j). Taken together these data paradoxically indicate that phosphorylation of α14 is not required for FRET activity yet kinase activities of RIPK1 and RIPK3 are nevertheless required for α14 FRET activation.

**Further refinement of α14 to generate SMART.** Although α14 monitored necroptosis, transient transfection of α14 blocked cell proliferation. To investigate the mechanisms how α14 monitors necroptosis and circumvent this drawback, we generated a further series of α14 mutants. To test whether binding of α14 to RIPK3 is required for monitoring necroptosis, we first mutated phenylalanine at position 234 of MLKL, that is critical for RIPK3 binding in vitro, to glutamic acid (F234E) (Fig. 2a). Unexpectedly, an F234E mutant of α14 still interacted with RIPK3 and showed the increase in the FRET/CFP ratio upon TZ stimulation (Fig. 2b, c). We next replaced amino acids at the indicated fragments with a set of four flexible Ser-Ala-Gly-Gly (SAGG) repeats to maintain the same spacing between Ypet and ECFP (Fig. 2a, Supplementary Fig. 2). The TZ-induced increase in the FRET/CFP ratio was partially or completely abolished in cells expressing Δα2α3, Δα3, or Δabc. In sharp contrast, TZ increased the FRET/CFP ratio of Δa and Δab comparable to that of α14 (Fig. 2b, Supplementary Fig. 3a, b). Notably, Δα2α3 did not interact with RIPK3 or show the increase in the FRET/CFP ratio (Fig. 2b, c). Three biosensors including α19, α59, and Δabc bound to RIPK3, but did not show the increase in the FRET/CFP ratio (Fig. 2c), suggesting that the interaction of RIPK3 with the FRET biosensors is prerequisite, but not sufficient for monitoring necroptosis. Among FRET biosensors showing the increase in the FRET/CFP ratio upon TZ stimulation, we were only able to obtain cells stably expressing Δab. Thus, this construct is henceforth referred to as SMART (a Sensor for MLKL activation by RIPK3 based on FRET).

Cotransfection experiments revealed that SMART and a mutant of SMART where all putatively phosphorylated serines and threonine were replaced with alanine (4ST5A), still interacted with RIPK3 in the absence or presence of GSK'872 (Fig. 2d). Thus, neither the kinase activity of RIPK3 nor the phosphorylation of SMART is required for their binding.

**SMART does not monitor apoptosis or necrosis.** To characterize the specificity of SMART further, we stably expressed SMART in L929 cells. Consistent with transient transfection assays, the FRET/CFP ratio rapidly increased in L929-SMART cells treated with TZ, but not TZ and GSK'872 (TZG) (Fig. 3a–c, Supplementary Movie 1, 2). We next treated L929-SMART cells with TG to induce apoptosis visualized by the rapid appearance of the cleaved form caspase 3 (Fig. 3d). Under these experimental conditions, the FRET/CFP ratio was not increased in TG-stimulated L929-SMART cells (Fig. 3c, e, Supplementary Movie 3). To test whether SMART monitors RIPK3-independent necrosis, we treated cells with an uncoupler, carbonyl cyanide m-chlorophenyl hydrazone (CCCP). As expected, CCCP-induced cell death was not blocked by either zVAD or GSK'872 (Fig. 3f), suggesting that these cells did not die by apoptosis or necroptosis. The FRET/CFP ratio was not increased in CCCP-stimulated L929-SMART cells and, on the contrary, rapidly decreased over time even prior to SYTOX positive staining indicating membrane permeabilization (Fig. 3c, g, Supplementary Movie 4). Together, SMART is a FRET biosensor that monitors necroptosis, but not apoptosis or necrosis.

**SMART monitors necroptosis in MEFs and aMoC1 cells.** To test whether SMART might be used generically to monitor necroptosis, we stably expressed SMART in murine embryonic fibroblast (MEF)s and a murine colonic epithelial cell line, aMoC1 cells. When these cells were treated with the combination of TNF, a SMAC mimetic, BV6, and zVAD (TBZ), they underwent necroptosis[23] (Supplementary Fig. 4a). As expected, TBZ increased the FRET/CFP ratio of SMART in both MEFs and aMoC1 cells (Supplementary Fig. 4b, c, Supplementary Movie 5, 6).

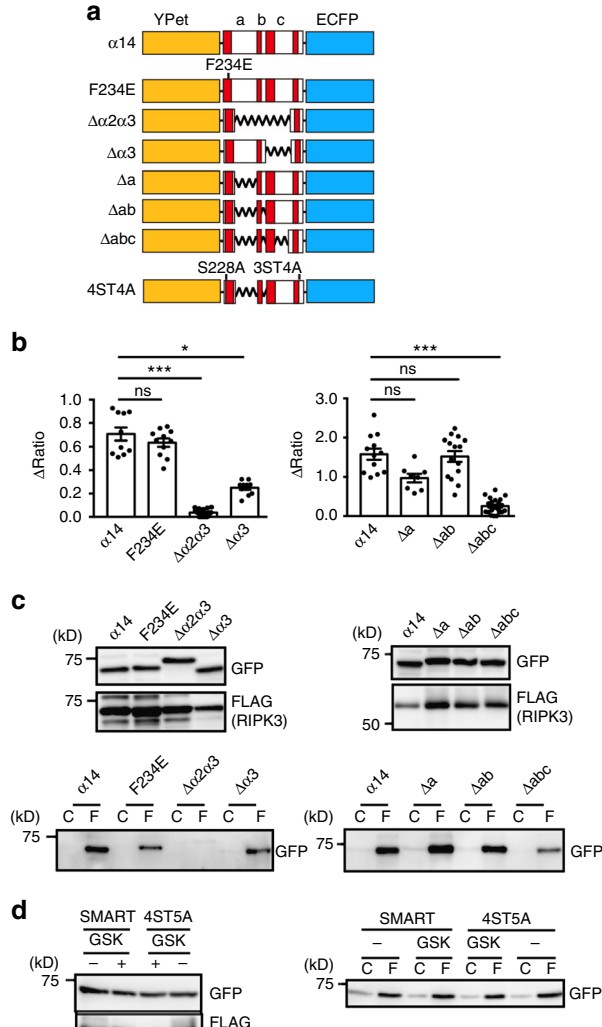

**Fig. 2** Interaction of α14 with RIPK3 is required for monitoring necroptosis. **a** Diagram of a series of α14 mutants. F234E; phenylalanine at 234 is substituted with glutamic acid, 4ST5A; S228, S345, T347, S349, and S352 were substituted with alanines. Zigzag lines indicate the regions that were replaced with SAGG repeats. Red boxes indicate each α helix. **b** L929 cells were transiently transfected with the indicated mutants and then stimulated with TZ, and the FRET/CFP ratio was analyzed as in Fig. 1b. Maximum changes of the FRET/CFP ratio. Each dot indicates individual cell ($n = 10$–14 cells). Statistical significance was determined using the one-way ANOVA test. *$P < 0.05$, ***$P < 0.001$, ns, not significant. **c** HEK293T cells were transiently transfected with the indicated mutants along with FLAG-RIPK3. Cell lysates were immunoprecipitated and analyzed as in Fig. 1c. Results are representative of two independent experiments. **d** GSK'872 treatment does not affect the binding of SMART to RIPK3. HEK293T cells were transiently transfected with SMART or 4ST5A along with FLAG-RIPK3. At 24 h after transfection, cells were untreated or treated with GSK'872 for 4 h, and cells were subjected to immunoprecipitation and analyzed as described in Fig. 1c. Results are representative of two independent experiments, or pooled results of two independent experiments (**b**). Error bars indicate s.e.m.

In sharp contrast, TBG induced apoptosis, but did not increase the FRET/CFP ratio of SMART in both MEFs and aMoC1 cells (Supplementary Fig. 4d, e). Thus, SMART monitors necroptosis, but not apoptosis in MEFs and aMoC1 cells as well as L929 cells.

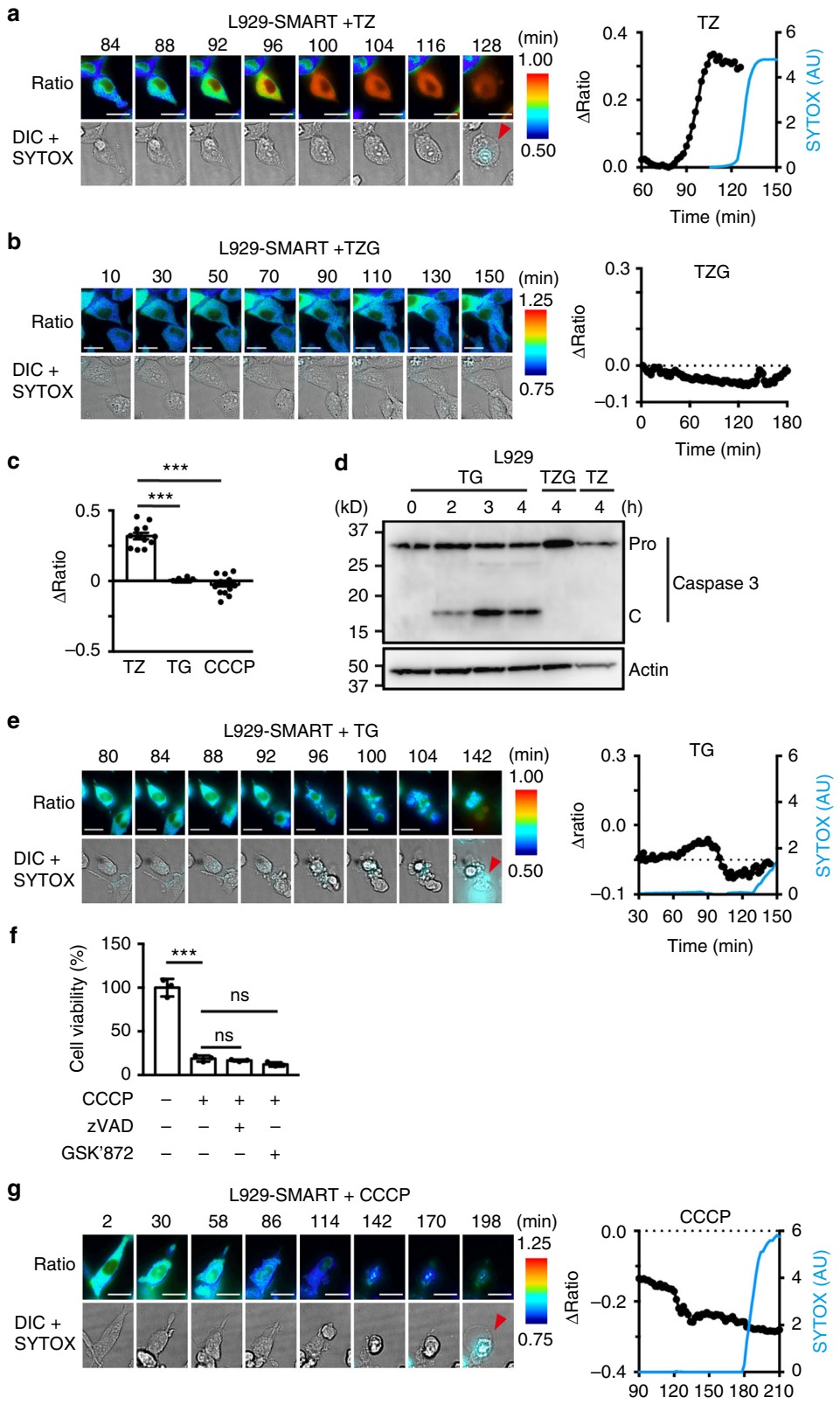

**Monitoring of necroptosis by SMART depends on RIPK3 and MLKL.** To further investigate the mechanisms how SMART monitors necroptosis, we knocked down the expression of *Ripk3* or *Mlkl* in L929-SMART cells. Treatment of cells with *Ripk3* or *Mlkl*, but not control siRNA blocked TZ-induced LDH release (Fig. 4a, b). As expected, knockdown of *Ripk3* abolished TZ-induced increase in the FRET/CFP ratio of SMART (Fig. 4c, Supplementary Fig. 5). TZ- and TBZ-induced increase in the

**Fig. 3** SMART does not monitor apoptosis or necrosis. **a**, **b**, **e**, **g** L929-SMART cells were treated with TZ (**a**), TZG (**b**), TG (**e**), or CCCP (**g**). The FRET/CFP ratio was analyzed as in Fig. 1b. Representative images of the ratio (left), and kinetics of ΔFRET/CFP ratio (black line) and relative intensities of SYTOX Orange (blue line) of a single cell (right) ($n$ = at least 10 cells). Relative intensities of SYTOX Orange in the nucleus were calculated at the indicated times, and are expressed as arbitrary units (AU). Ratio and DIC + SYTOX indicate the FRET/CFP ratio, and merged images of DIC and SYTOX Orange, respectively. Red arrowheads indicate SYTOX-positive cells. Scale bars, 20 μm. Color scales indicate pseudocolor images of the FRET/CFP ratio. **c** Maximum changes of the FRET/CFP ratio. Each dot indicates individual cell ($n$ = 10 cells). Statistical significance was determined using the one-way ANOVA test. ***$P < 0.001$. Error bars indicate s.e.m. **d** L929 cells were stimulated as indicated for the indicated time periods. Cell lysates were analyzed by immunoblotting with the indicated antibodies. Pro and C indicate the proform and cleaved form of caspase 3, respectively. **f** L929 cells were stimulated with CCCP in the absence or presence of zVAD or GSK'872 for 2 h, and cell viability was determined by WST assay. Results are mean ± s.d. of triplicate samples. Statistical significance was determined using the one-way ANOVA test. **$P < 0.01$, ns, not significant. All results are representative of two or three independent experiments

FRET/CFP ratio was also abolished in L929-SMART cells treated with *Mlkl* siRNA and *Mlkl*−/− MEFs-SMART, respectively (Fig. 4c–e, Supplementary Fig. 5). Moreover, the kinetics of the increase in the FRET/CFP ratio of SMART correlated with those of phosphorylation of RIPK1, RIPK3, and MLKL in L929 cells and wild-type (WT) MEFs following TBZ stimulation (Fig. 4f–i). These results suggest that the increase in the FRET/CFP ratio of SMART depends on endogenous RIPK3 and MLKL, and is tightly correlated with hallmarks of necroptosis.

**SMART monitors oligomerization of MLKL**. RIPK3 kinase activity and endogenous MLKL increased the FRET/CFP ratio of SMART (Figs. 3b and 4a–d), suggesting that SMART monitors the oligomerization of endogenous MLKL. To test this possibility, we transfected *Mlkl*−/− MEFs-SMART with doxycycline (Dox)-inducible lentiviral vectors for WT MLKL, MLKL L280P, or MLKL Q343A. We previously reported that MLKL Q343A is an auto-activated form of MLKL that spontaneously forms oligomers, resulting in necroptosis, whereas MLKL L280P blocks TBZ-induced necroptosis[24]. We verified Dox-induced expression of WT MLKL, MLKL L280P, or MLKL Q343A in *Mlkl*−/− MEFs (Fig. 5a) and, as expected, the expression of WT MLKL rendered *Mlkl*−/− MEFs susceptible to TBZ-induced necroptosis (Fig. 5b). Furthermore, as we previously reported, Dox-induced expression of MLKL Q343A resulted in necroptosis of *Mlkl*−/− MEFs-SMART (Fig. 5b). More importantly, the expression of MLKL Q343A induced its oligomers and increased the FRET/CFP ratio of SMART (Fig. 5c–e). In sharp contrast, both TBZ did not induce necroptosis or increase the FRET/CFP ratio of SMART in *Mlkl*−/− MEFs-SMART reconstituted with MLKL L280P (Fig. 5b, d, e). Together these data support our hypothesis that SMART monitors the oligomerization of MLKL and/or subsequent plasma membrane translocation of oligomerized MLKL even in the absence of TNF stimulation.

**Generation of human version of SMART**. As the murine α14 SMART prototype did not work in human cells (Supplementary Fig. 6), we generated a human version of SMART (hSMART). hSMART was composed of helices α1 to α5 of the KL domain of hMLKL, in which the a and b regions corresponding to mMLKL were replaced with SAGG repeats (Supplementary Fig. 2c, Fig. 6a). hSMART efficiently monitored necroptosis in a human colon cancer cell line, HT29, following the induction of necroptosis (TBZ), but not apoptosis (TBG) (Fig. 6b, c, e). Execution of necroptosis by MLKL is composed of sequential two steps: oligomerization of phosphorylated MLKL and plasma membrane translocation of oligomerized MLKL. To address which steps are crucial for the increase in the FRET/CFP ratio of SMART, we stimulated HT29-SMART cells with TBZ in the absence or presence of Necrosulfonamide (NSA). NSA targets cysteine at 86 (C86) of the N-terminal domain (NTD) of human, but not

murine MLKL, and inhibits necroptosis through preventing plasma membrane translocation of oligomerized MLKL[25,26]. As expected, NSA blocked TBZ-induced plasma membrane translocation of oligomerized MLKL, but not oligomerization of MLKL itself (Fig. 6f). Although hSMART does not possess C86, NSA treatment nevertheless abolished the increase in the FRET/CFP ratio following TBZ stimulation (Fig. 6d, e). Since hSMART did not translocate to membrane fraction under these experimental conditions (Fig. 6f), direct interaction of hSMART with oligomerized MLKL is not required for the increase in the FRET/CFP ratio of hSMART.

**SMART monitors poly(I:C)-induced necroptosis**. A previous study reported that poly(I:C) + BZ (PBZ) induces necroptosis in a human keratinocyte cell line, HaCaT cells[27]. We found that hSMART also monitored PBZ-induced necroptosis in HaCaT cells, but not PBG-induced apoptosis (Fig. 7a–d). As poly(I:C) stimulation induces necroptosis through TLR3[27,28], SMART monitors both TLR3-induced and TNFR1-induced necroptosis.

**Two sequential steps of HMGB1 release during necroptosis**. While DAMPs, such as HMGB1 and histones, are considered to be released from dead cells due to the collapse of both the nuclear and cytoplasmic membrane integrity[9,10], the underlying mechanisms are not fully understood. To address this issue, we investigated the kinetics of the release of various DAMPs along with the progression of cell death. To visualize nuclear DAMPs, we fused mCherry to the C-terminal portion of HMGB1 and histone H3, resulting in the generation of HMGB1-mCherry and histone H3-mCherry, respectively. We first confirmed that the kinetics of the release of HMGB1-mCherry was comparable to that of endogenous HMGB1 from L929 cells stably expressing HMGB1-mCherry cells stimulated with TZ (Supplementary Fig. 7a). We next transfected L929-SMART cells with HMGB1-mCherry or histone H3-mCherry. As expected, HMGB1-mCherry and histone H3-mCherry were localized in the nucleus (Fig. 8a, b). HMGB1-mCherry abruptly disappeared between 2 and 3 h after TZ stimulation and this happened before the nucleus became positive for SYTOX (Fig. 8a, c). However, in the same time-frame, the histone H3-mCherry remained constant in the nucleus even after the nucleus became positive for SYTOX (Fig. 8b). The FRET/CFP ratio of SMART gradually increased before the signals of HMGB1-mCherry disappeared (Fig. 8c). Loss of HMGB1 from the nucleus occurred quickly and corresponded with a precipitate rise in cytosolic HMGB1 levels (Fig. 8c, d). However, this increase in cytosolic HMGB1 was short-lived reflecting rapid release of HMGB1 into the extracellular space (Fig. 8c, d, Supplementary Movie 7). These results suggest that the integrity of the nuclear membrane was already disrupted before the cytoplasmic membrane became ruptured. Western blotting analysis showed that the release of endogenous

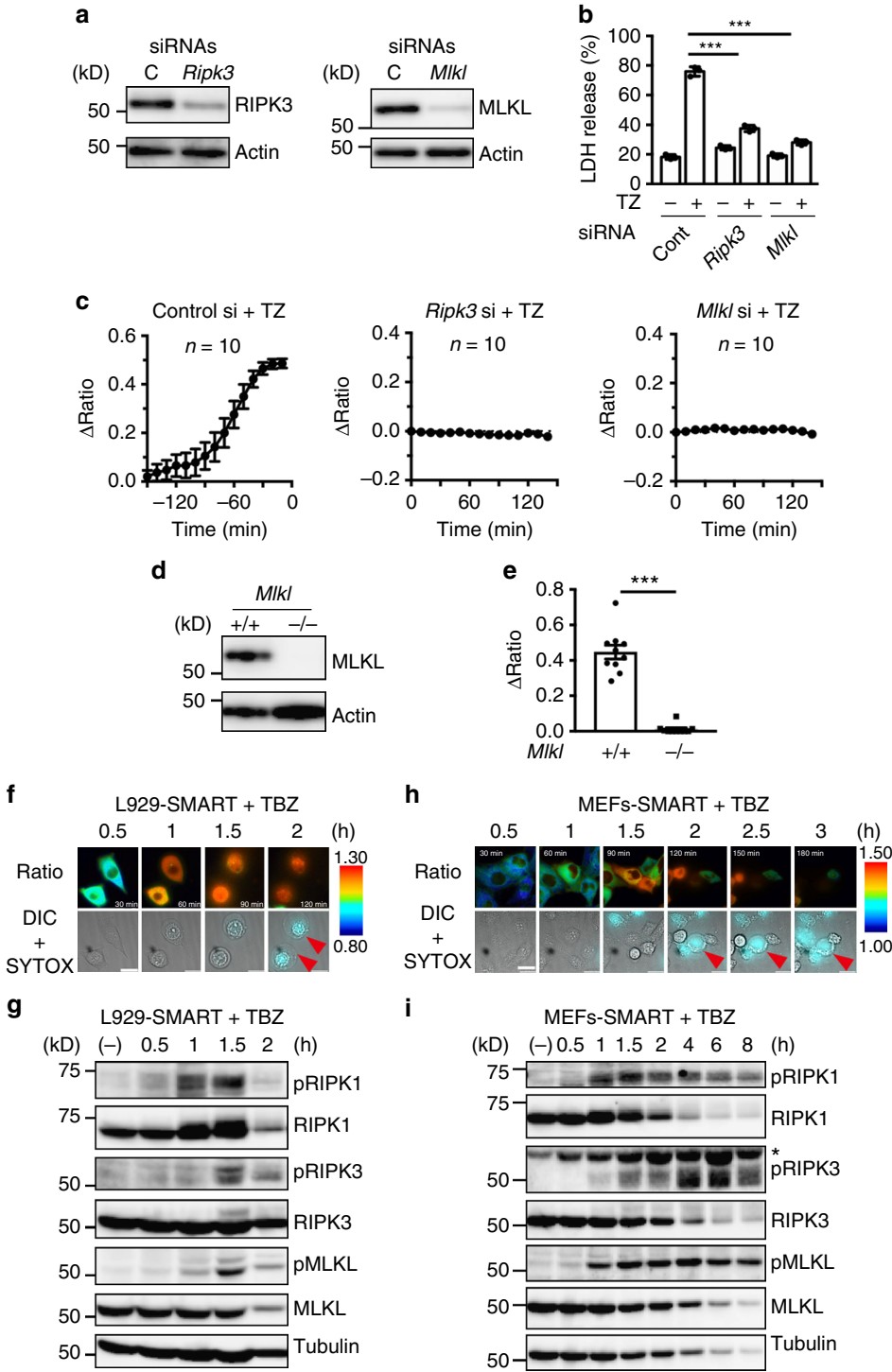

HMGB1 in the culture supernatant started approximately 2 h after a necroptotic stimulus, and as previously reported, was almost undetectable in the culture supernatant of cells undergoing apoptosis (Supplementary Fig. 7a, b). Similar findings were obtained by MEFs and aMoC1 cells (Supplementary Fig. 7c, d). The release of HMGB1 was relatively slow in HT29 and HaCaT cells compared to murine cell lines (Supplementary Fig. 7e, f). On the other hand, histone H3 was not, or marginally released into the extracellular spaces from cells died by either apoptosis or necroptosis at least under our experimental conditions (Supplementary Fig. 7b-f).

**LCI-S uncovers two different modes of HMGB1 release**. To further investigate the kinetics of the release of HMGB1 and execution of necroptosis, we visualized the extracellular release of HMGB1 by LCI-S in combination with SMART at the single cell level. Our preliminary experiments revealed that commercially available antibodies against HMGB1 were not suitable for sandwich fluoro-immunoassay under physiological conditions. We plated a single cell of L929 co-expressing SMART and HMGB1-mCherry on a microfabricated-well array chip that was precoated with the anti-mCherry antibody. Simultaneous live imaging of SMART, intracellular HMGB1-mCherry, and extracellularly

**Fig. 4** Monitoring of necroptosis by SMART depends on RIPK3 and MLKL. **a, b** Knockdown of *Ripk3* or *Mlkl* abolishes the TZ-induced increase in the FRET/CFP ratio of SMART. L929-SMART cells were transfected with control, *Ripk3*, or *Mlkl* siRNAs. Expression of RIPK3 or MLKL was analyzed by immunoblotting with the indicated antibodies (**a**). After transfection, cells were unstimulated or stimulated with TZ for 8 h. Cell viability was determined by LDH release assay (**b**). Results are mean ± s.d. of triplicate samples. Statistical significance was determined using the one-way ANOVA test. ***$P < 0.001$. **c** L929-SMART cells were treated with the indicated siRNAs and then stimulated with TZ. The FRET/CFP ratio was analyzed as in Fig. 1b. Kinetics of average ΔFRET/CFP ratio of cells ($n = 10$ cells). Time 0 for control siRNA indicates the time when cells underwent rupture, whereas the time for *Ripk3* or *Mlkl* siRNAs indicates the time after stimulation. **d, e** The TZ-induced increase in the FRET/CFP ratio of SMART is abolished in *Mlkl*−/− MEFs. Expression of MLKL in *Mlkl*−/− MEFs was analyzed by immunoblotting (**d**). Wild-type and *Mlkl*−/− MEFs stably expressing SMART were stimulated with TBZ, and the FRET/CFP ratio was analyzed as in Fig. 1b. Maximum changes of the FRET/CFP ratio (**e**). Each dot indicates individual cell ($n = 10$ cells). Statistical significance was determined using the unpaired two-tailed Student's *t* test. ***$P < 0.001$. **f, h** L929-SMART cells and MEFs-SMART were stimulated with TBZ and the FRET/CFP ratio was analyzed as in Fig. 1b. Representative images of the ratio of a single cell (left) ($n = 10$ cells). Ratio and DIC + SYTOX indicate the FRET/CFP ratio, and merged images of DIC and SYTOX Orange, respectively. Red arrowheads indicate SYTOX-positive cells. Scale bars, 20 μm. Color scales indicate pseudocolor images of the FRET/CFP ratio. **g, i** L929-SMART cells and MEFs-SMART were stimulated with TBZ for the indicated times and cell lysates were analyzed by immunoblotting with the indicated antibodies. All results are representative of two independent experiments. Error bars indicate s.e.m. (**c, e**)

---

released HMGB1-mCherry by LCI-S further substantiated that the FRET/CFP ratio started to increase before the signals of HMGB1-mCherry disappeared in the cell (Fig. 9a, b, Supplementary Movie 8). Given that there was a time lag between the disappearance of the signals of HMGB1-mCherry in the nucleus and detection of extracellularly released HMGB1-mCherry by LCI-S (Fig. 9a, b), these results further substantiated that HMGB1-mCherry was released from the nucleus before plasma membrane ruptured. Average time from an increase in the FRET/CFP ratio to the extracellular release of HMBG1-mCherry was approximately 43.8 min (Fig. 9c).

Intriguingly, we found that in some cells, HMGB1-mCherry release was stopped within 10 min, whereas in other cells, HMGB1-mCherry release lasted more than 100 min (Fig. 9d–f, Supplementary Movie 9, 10). Thus, we surmised the release mode of HMGB1-mCherry could be divided into two groups: one is a burst-mode and the other is a sustained-mode. To compare these two modes quantitatively, we estimated the duration of the HMGB1-mCherry release in individual cells and classified them into two groups by k-means clustering. The representative duration of extracellular HMGB1 release of the burst-mode and the sustained-mode are 7.1 and 109 min, respectively (Fig. 9f). To investigate the kinetics of the extracellular release of HMGB1-mCherry in more detail, we quantified the relative intensities of intracellular (total), extracellular (TIRF), and nuclear HMGB1-mCherry in a single cell, respectively. Then, we plotted representative kinetics of signal intensities of a single cell at the indicated times before and after the extracellular release of HMGB1-mCherry (Fig. 9g). The intracellular and nuclear signals of HMGB1-mCherry drastically diminished in burst-mode cells, but remained relatively high in sustained-mode cells after the cytoplasmic membrane rupture (Fig. 9g, left vs right). Consistently, some portions of nuclear HMGB1-mCherry still existed in the nucleus of a sustained-mode cell, whereas most nuclear HMGB1-mCherry disappeared in a burst-mode cell (Epi, Fig. 9d). Together, the extent of both nuclear and cytoplasmic membrane damage induced by MLKL might critically determine whether cells release HMGB1 in a burst-mode or a sustained-mode.

**CHMP4B is involved in a sustained-mode release of HMGB1.** The endosomal sorting complex required for transport (ESCRT) is involved in mediating receptor sorting, membrane remodeling, and membrane scission[29]. A recent study has shown that MLKL binds to the ESCRT proteins and generates extracellular vesicle[30]. Moreover, CHMP2A and CHMP4B, components of the ESCRT-III complex, have been shown to delay or prevent TNF-induced necroptosis through shedding MLKL-containing vesicles[31]. Taken that the knockdown of *Chmp2a* or *Chmp4b* enhances

TNF-induced necroptosis[31], we surmised that the ESCRT-III proteins maintained a sustained-mode release of HMGB1 by promoting membrane repair. To test this possibility, we knocked down *Chmp4b* in L929-SMART/HMGB1-mCherry cells by siRNA (Fig. 10a). After TZ stimulation, we monitored HMGB1-mCherry release by LCI-S and estimated the duration of the release of HMGB1 of individual cell. Intriguingly, knockdown of *Chmp4b* substantially reduced the duration of the HMGB1-mCherry release compared to control siRNA-treated cells (Fig. 10b). Moreover, when we classified the assembly from both of these siRNA-treated cells into two groups based on the duration of the HMGB1-mCherry release by k-means clustering, cells that released HMGB1-mCherry via the sustained-mode were abolished in *Chmp4b*, but not in control siRNA-treated cell populations (Fig. 10b, Supplementary Fig. 8).

As expected, the time between the start of the release of HMGB1 and the burst of cells was shortened, and ΔFRET/CFP ratio was more rapidly increased in cells treated with *Chmp4b* siRNA than those with control siRNA (Fig. 10c, d). Together, these results suggest that CHMP4B contributes to maintain a sustained-mode of HMGB1 release, possibly by promoting plasma membrane repair.

**Discussion**

In the present study, we developed a FRET biosensor that detected necroptosis in living cells. The increase in the FRET/CFP ratio of SMART depended on RIPK3 and MLKL, and was correlated with phosphorylation of RIPK3 and MLKL, hallmarks of necroptosis. Moreover, SMART monitored plasma membrane translocation of oligomerized MLKL even in the absence of TNF stimulation. SMART monitored necroptosis, but not apoptosis or necrosis. Simultaneous live imaging of SMART and the release of nuclear DAMPs by LCI-S uncovered two different modes of the release of HMGB1 from cells undergoing necroptosis. Moreover, CHMP4B, a component of the ESCRT-III complex might determine whether a cell exhibits a burst-mode or a sustained-mode of HMGB1 release.

Many groups including us developed FRET biosensors to monitor apoptosis in living cells[16,18,32–34]. Imaging of necroptosis is rather difficult, since there has been no specific polypeptide(s) that are cleaved by protease(s) activated during necroptosis. Taken that the phosphorylation of MLKL is prerequisite for necroptosis[21,22], we assumed that a short fragment containing a MLKL-phosphorylation site (~15 amino acids) and a phosphopeptide recognition domain, such as the Forkhead-associated (FHA) domain fused by a linker peptide might be suitable for generating a FRET biosensor to monitor necroptosis. However, such fragment could not monitor necroptosis. Thus, we next

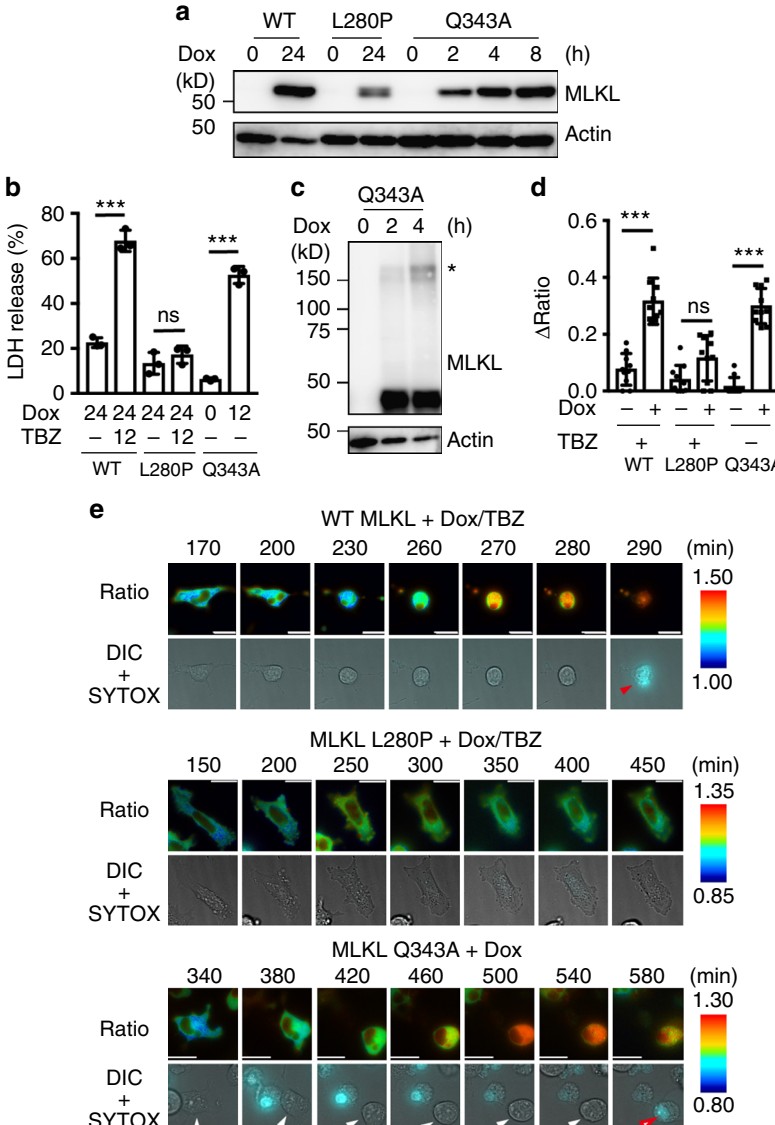

**Fig. 5** SMART monitors oligomerization of MLKL. **a** *Mlkl*−/− MEFs-SMART were transfected with Dox-inducible lentiviral vectors for WT MLKL, MLKL L280P, or MLKL Q343A. Cells were treated with Dox for the indicated times. Expression of MLKL mutants was analyzed by immunoblotting with the anti-MLKL antibody. **b** *Mlkl*−/− MEFs-SMART/WT MLKL and MLKL L280P were treated with Dox as in (**a**) and then untreated or further stimulated with TBZ for 12 h. *Mlkl*−/− MEFs-SMART/MLKL Q343A were treated with Dox alone for 12 h. Cell viability was determined by LDH release assay. Results are mean ± s.d. of triplicate samples. Statistical significance was determined using the unpaired two-tailed Student-*t* test. ***$P < 0.001$, ns, not significant. **c** *Mlkl*−/− MEFs-SMART/MLKL Q343A were treated with Dox for the indicated times and cell lysates were subjected to SDS-PAGE under non-reducing conditions. Oligomers of MLKL were analyzed by immunoblotting with the anti-MLKL antibody. Asterisk indicates oligomers. **d** Cells were treated as in (**b**) and the FRET/CFP ratio was analyzed as in Fig. 1b. Maximum changes of the FRET/CFP ratio. Each dot indicates individual cell ($n = 10$ cells). Statistical significance was determined using the unpaired two-tailed Student-*t* test. ***$P < 0.001$, ns, not significant. Error bars indicate s.e.m. **e** Cells were treated and analyzed as in (**b**). Representative images of the ratio of a single cell expressing SMART ($n = 10$ cells). Ratio and DIC + SYTOX indicate the FRET/CFP ratio and merged images of DIC and SYTOX Orange, respectively. Red and white arrowheads indicate SYTOX-positive cells and cells undergoing necroptosis, respectively. Scale bars, 20 μm. Color scales indicate pseudocolor images of the FRET/CFP ratio. All results are representative of two independent experiments

focused on conformational changes of MLKL induced by RIPK3 binding[19,24]. Consistent with our hypothesis, one of the FRET probes containing the α14 fragments of MLKL allowed us to monitor necroptosis. Unexpectedly, the quadruple mutant 3ST4A of α14 and SMART 4ST5A did monitor necroptosis after TZ stimulation. In sharp contrast, GSK'872 suppressed TZ-induced necroptosis and the increase in the FRET/CFP ratio of SMART. Moreover, TZ- or TBZ-induced increase in the FRET/CFP ratio of SMART was abolished in L929 cells treated with *Ripk3* or *Mlkl* siRNAs and *Mlkl*−/− MEFs. Together, these results suggest that

the phosphorylation of endogenous MLKL, but not the phosphorylation of SMART probe itself is indispensable for the increase in the FRET/CFP ratio of SMART. Expression of a constitutive active mutant, MLKL Q343A that spontaneously forms oligomers and induced necroptosis, resulted in the increase in the FRET/CFP ratio of SMART even in the absence of TNF stimulation. NSA suppressed TBZ-induced increase in the FRET/CFP ratio of hSMART in HT29 cells, suggesting that SMART monitors plasma membrane translocation of oligomerized MLKL. At this moment, the detailed molecular mechanisms how SMART

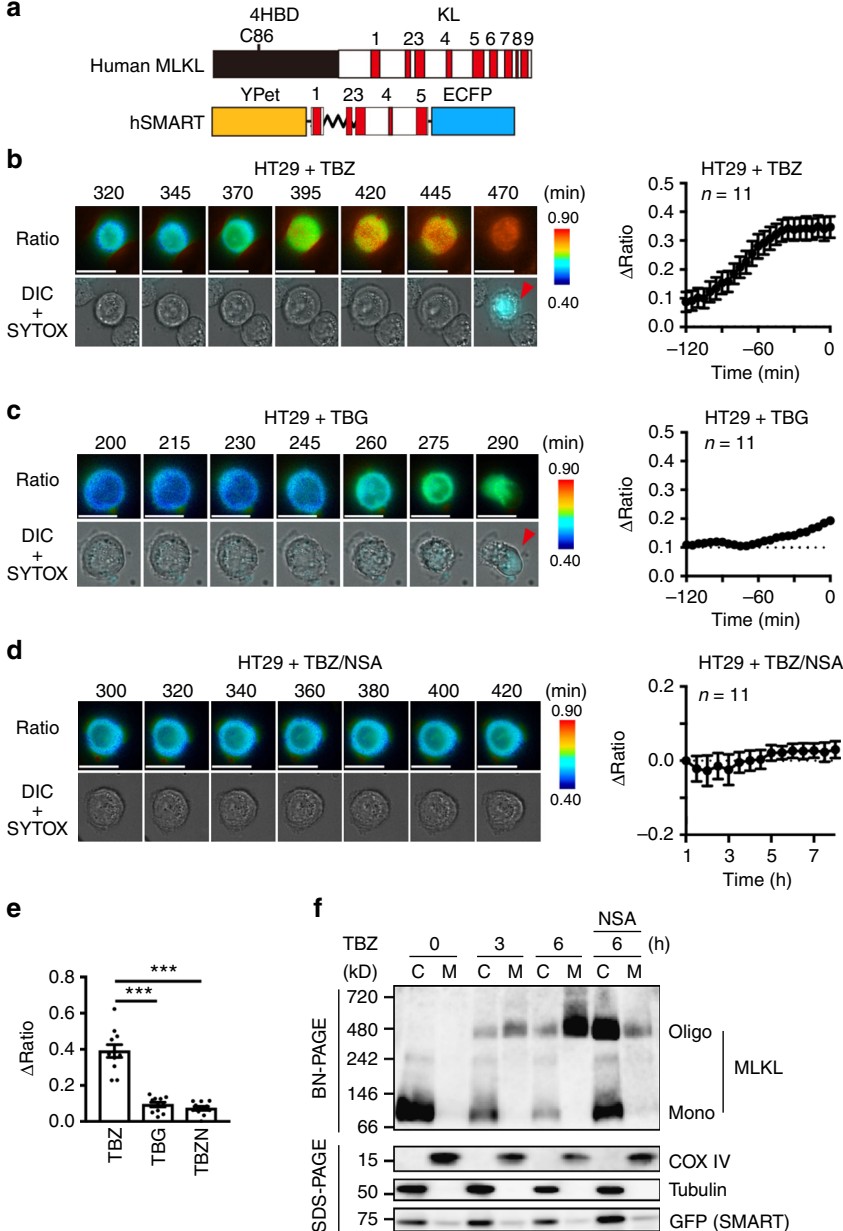

**Fig. 6** hSMART monitors necroptosis in human cells. **a** Diagram of hSMART. **b**–**e** HT29 cells stably expressing hSMART were stimulated with TBZ (**b**), TBG (**c**), or TBZ + NSA (**d**). The FRET/CFP ratio was analyzed as in Fig. 1b. Representative images of the ratio of a single cell (left) and kinetics of average of ΔFRET/CFP ratio of cells (right) (n = 11 cells). Ratio and DIC + SYTOX indicate the FRET/CFP ratio, and merged images of DIC and SYTOX Orange, respectively. Red arrowheads indicate SYTOX-positive cells. Time 0 indicates the time of cells that became SYTOX-positive (**b**, **c**). Otherwise indicated, time indicates the time after stimulation (**d**). Scale bars, 20 μm. Color scales indicate pseudocolor images of the FRET/CFP ratio. Maximum changes of the FRET/CFP ratio (**e**). Each dot indicates individual cell (n = 11 cells). Statistical significance was determined using the one-way ANOVA test. ***P < 0.001. Error bars indicate s.e.m. Pooled results of two independent experiments. **f** SMART monitors plasma membrane translocation of oligomerized MLKL. HT29-SMART cells were stimulated with TBZ in the absence or presence of NSA for the indicated times. Then cytosolic and membrane fractions were prepared and subjected to BN-PAGE (upper panel) or SDS-PAGE under reducing conditions (lower panel). Expression of each protein in cytosolic (C) or membrane (M) fractions was determined by immunoblotting with the indicated antibodies. Oligo and mono indicate oligomers and monomer of MLKL, respectively. Results are representative of two independent experiments

senses plasma membrane translocation of oligomerized MLKL are currently unknown. Since SMART did not translocate to the plasma membrane in cells undergoing necroptosis, the increase in the FRET/CFP ratio of SMART does not appear to be mediated by direct interaction with oligomerized MLKL on the plasma membrane. One of the plausible explanations would be that SMART might monitor drastic changes of cellular conditions induced by plasma membrane translocation of oligomerized

MLKL, such as relative ratios of RIPK3 and MLKL in the cytosol. Further study will be required to address this issue.

Time-lapse imaging of the release of nuclear DAMPs and SMART uncovered differential regulation of the release of HMGB1 and histone H3 from cells undergoing necroptosis or apoptosis. HMGB1 was rapidly released from necroptotic cells just after cells became positive for SYTOX. It is currently unknown why HMGB1 bound to chromatin was rapidly released

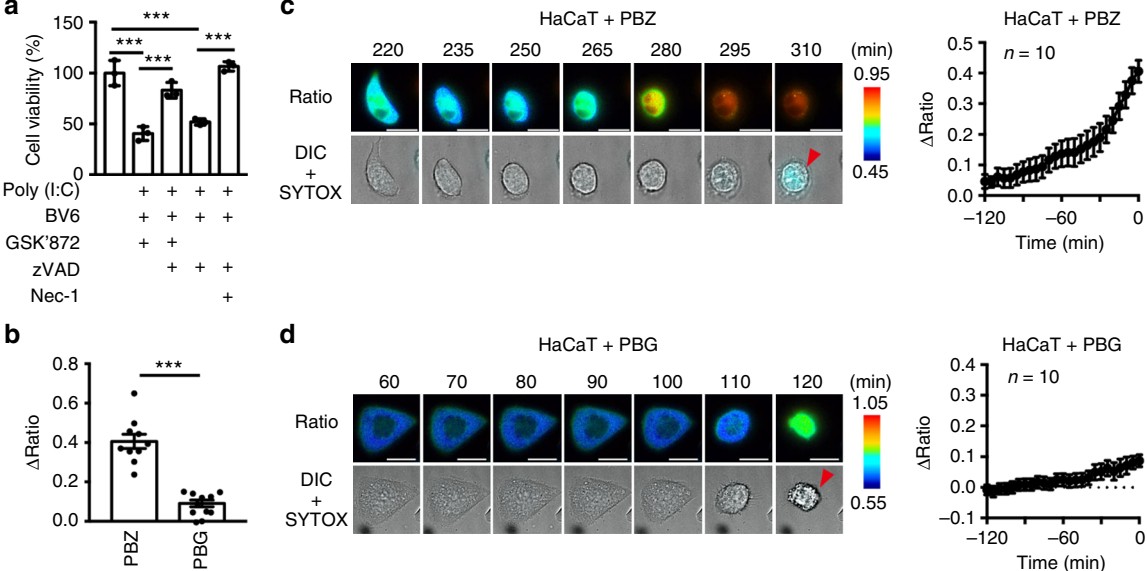

**Fig. 7** SMART monitors poly(I:C)-induced necroptosis. **a** HaCaT cells were stimulated with the combination of the indicated agents for 24 h. Cell viability was determined by WST assay. Results are mean ± s.d. of triplicates. Statistical significance was determined using the one-way ANOVA test. ***$P < 0.001$. Results are representative of two independent experiments. **b–d** HaCaT cells stably expressing hSMART were stimulated with Poly(I:C) + BV6 + zVAD (PBZ) or Poly(I:C) + BV6 + GSK'872 (PBG). The FRET/CFP ratio was analyzed as in Fig. 1b. Maximum changes of the FRET/CFP ratio (**b**). Each dot indicates individual cell ($n = 10$ cells). Statistical significance was determined using the unpaired two-tailed Student-$t$ test. ***$P < 0.001$. Representative images of the ratio (left) and kinetics of ΔFRET/CFP (right) of a single cell ($n = 10$ cells) (**c**, **d**). Red arrowheads indicate SYTOX-positive cells. Time 0 indicates the time of cells that became SYTOX-positive. Scale bar, 20 μm. Color scales indicate pseudocolor images of the FRET/CFP ratio. Error bars indicate s.e.m. Pooled results are two independent experiments

from the nucleus into the extracellular spaces. Disulfide-HMGB1, but not the reduced form of HMGB1, is released from cells during necroptosis[35]. Thus, we surmise that conformation change of HMGB1 induced by disulfide bonding might weaken the interaction of HMGB1 with chromatin, resulting in rapid release from the nucleus to the extracellular spaces in necroptotic cells. Taken that disulfide-HMGB1, but not reduced HMGB1 has a potent activity to produce inflammatory cytokines[35,36], only biologically active disulfide-HMGB1 might be released from necroptotic cells. Analysis of the detailed kinetics of the release of HMGB1 revealed that the release of HMGB1 from the nucleus to extracellular space appeared to consist of two sequential steps: the first step is cytoplasmic translocation from the nucleus, and the second step is extracellular release from the cytoplasm. This suggests that nuclear membrane damage is induced during the execution of necroptosis before the cytoplasmic membrane becomes ruptured. Taken that oligomers of MLKL might translocate to lipid bilayers of any organelles in cells including nuclear membrane[6,7,37,38], it is reasonable to hypothesize that nuclear membrane might be disrupted before the breakdown of the cytoplasmic membrane.

LCI-S uncovered that there were two different modes of HMGB1 release, "a burst-mode" and "a sustained-mode". The ESCRT complex has been shown to be involved in plasma membrane wound repair[29,39]. Moreover, recent studies reported that components of the ESCRT-III protein complex are involved in attenuation of TNF-induced necroptosis through extrusion of MLKL-containing vesicles[30,31]. Taken that the knockdown of *Chmp4b* abolished cells showing a sustained-mode release of HMGB1, the balance between the extent of pore forming activity of MLKL and membrane repairing capacity of proteins of the ESCRT complex, such as CHMP4B, might determine whether cells release HMGB1 via the sustained-mode or burst-mode. It would be intrigued to investigate mRNA levels of *Mlkl* and *Chmp4b* of each cell that exhibits a sustained-mode or a burst-mode.

Currently, the exact biological significances of these two different modes of the releases of DAMPs remain unclear. It is reasonable to surmise that a burst-mode and a sustained-mode cell might elicit qualitatively or quantitatively different biological responses of neighboring cells that sense DAMPs, such as macrophages and dendritic cells (DCs). Thus, it would be interesting to test whether amounts of production of inflammatory cytokines by macrophages or DCs might be different when cells were cocultured with a burst-mode cell or a sustained-mode cell. Alternatively, taken that HMGB1 also has a chemoattractant activity[40,41], a burst-mode cell might recruit cells such as DCs or neutrophils from a distant site to inflammation sites where cells undergo necroptosis by a concentration gradient of HMGB1 like a chemokine.

RIPK3-dependent oligomerization and subsequent translocation of MLKL to the cytoplasmic and nuclear membrane are reminiscent of caspase 1- or 11-dependent cleavage of Gasdermin (GSDM) D and its subsequent translocation to the cytoplasmic membrane[38,42,43]. Very recently, two groups have reported that caspase 3-dependent cleavage of GSDME/DFNA5 mediates secondary necrosis[44,45]. So far, it is unclear whether GSDMD or GSDME translocate to the nuclear membrane and participates in the release of nuclear DAMPs. In this respect, simultaneous live imaging of nuclear DAMPs, LCI-S, and FRET biosensors that monitor cleavages of the GSDM family will address whether GSDMD or GSDME are also involved in the collapse of the nuclear membrane and the release of nuclear DAMPs.

Consistent with the results of HMGB1, IL-1α and IL-33 are nuclear proteins that do not possess signal peptides and released from damaged cells[8,9]. Our preliminary experiments showed that kinetics of the release of IL-33 from cells undergoing necroptosis appeared to be different from that of HMGB1. Thus, SMART in combination with LCI-S might further increase the understanding of mechanisms of necroptosis and the release of DAMPs.

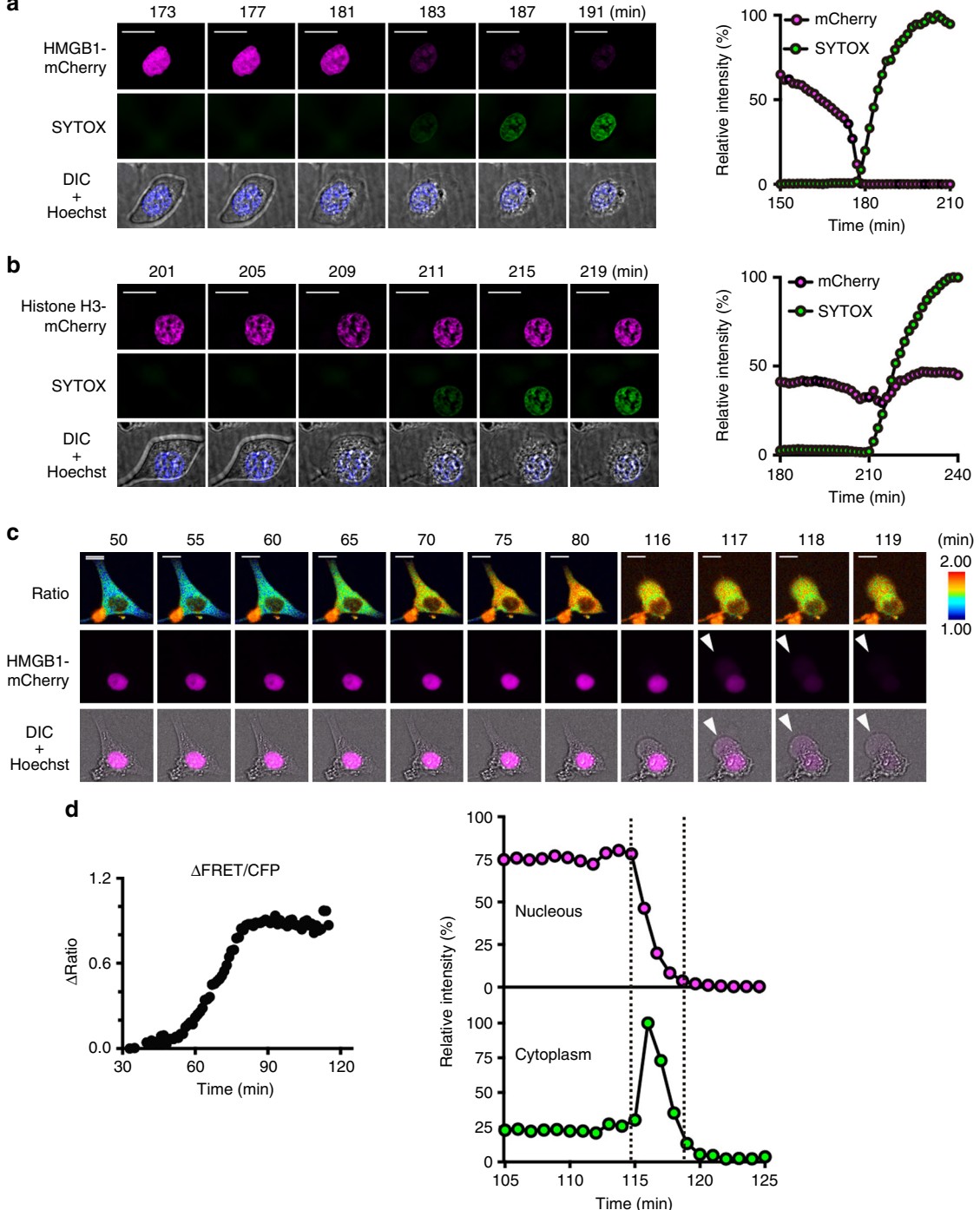

**Fig. 8** Sequential two-step release of HMGB1 from cells undergoing necroptosis. **a**, **b** L929 cells stably expressing HMGB1-mCherry or histone H3-mCherry were stimulated with TZ. The signals of HMGB1- or histone H3-mCherry were analyzed every 2 min. Representative images of HMGB1-mCherry (**a**) or histone H3-mCherry (**b**) (upper), SYTOX Green (middle), and merged images of DIC and Hoechst (lower) (left) (n = 5 cells). Scale bars, 20 μm. Intensities of mCherry (magenta) and SYTOX Green (green) were quantified and relative intensities were plotted at the indicated times (right). **c**, **d** L929-SMART cells were transiently transfected with HMGB1-mCherry and stimulated with TZ. The FRET/CFP ratio was analyzed as in Fig. 1b. Representative images of the FRET/CFP ratio (upper), HMGB1-mCherry (middle), and merged images of DIC and HMGB1-mCherry (lower) (n = 5 cells) (**c**). White arrowheads indicate released HMGB1-mCherry in the cytoplasm. Scale bars, 20 μm. ΔFRET/CFP ratio (left), and relative intensity of HMGB1-mCherry in the nucleus and the cytoplasm (right) were plotted at the indicated times after stimulation (**d**). Intensities of HMGB1-mCherry in the cytoplasm were determined by subtracting nuclear intensities of HMGB1-mCherry from intracellular (total) intensities of HMGB1-mCherry. Two vertical dotted lines indicate the times when nuclear efflux of HMGB1-mCherry started and extracellular release of HMGB1-mCherry was terminated, respectively. All results are representative of at least three independent experiments

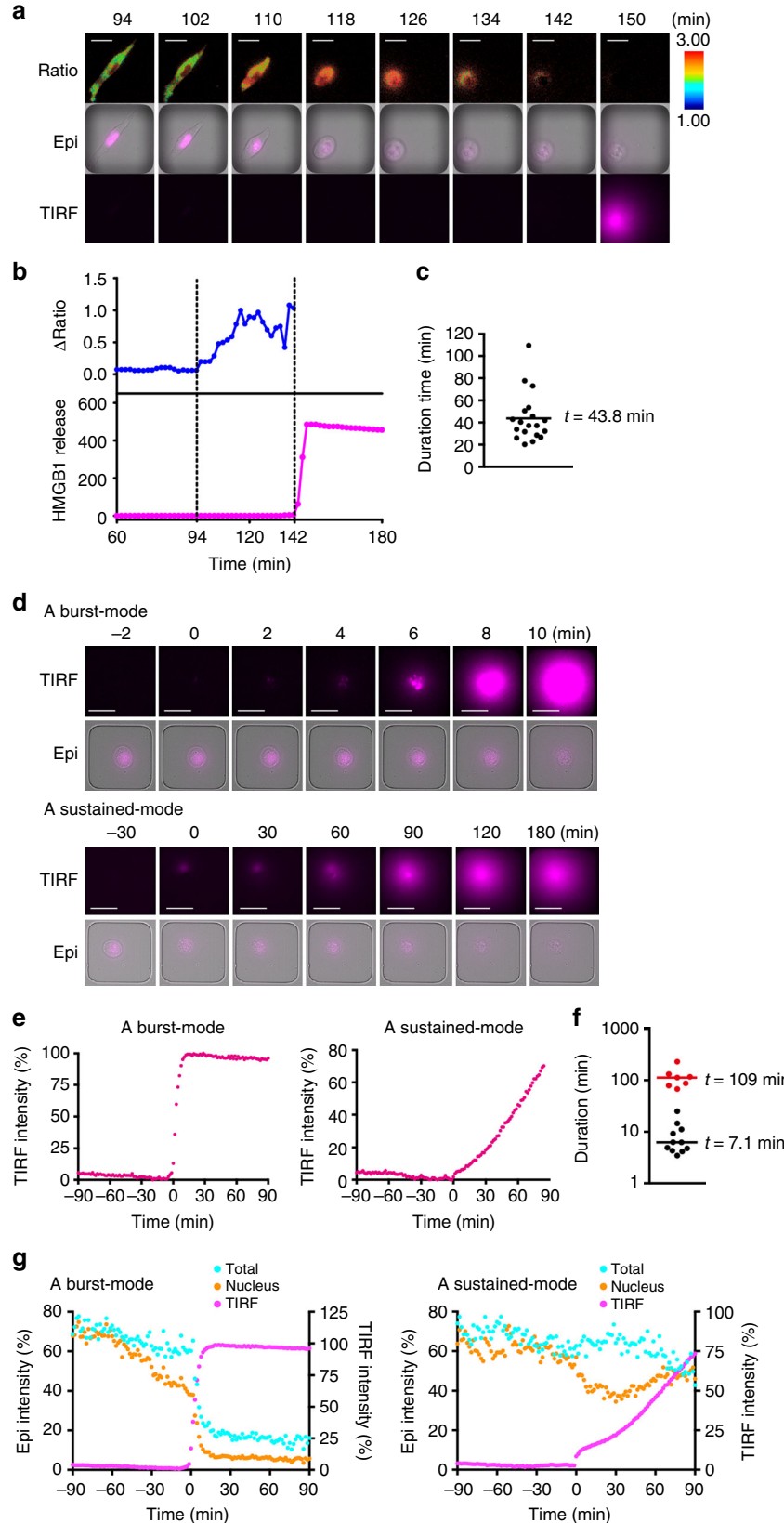

## Methods

**Reagents**. Murine TNF (34-8321, eBioscience), human TNF (BMS301, eBioscience), poly(I:C) (ALX-746-021, Enzo Life Sciences), Birinapant (CT-BIRI, Tetralogic Pharmaceuticals), BV6 (B4653, ApexBio), CCCP (CAS 555-60-2, Calbiochem), GSK'872 (530389, Merck), Hoechst 33342 (H3570, ThermoFisher Scientific), Nec-1 (N9037, Sigma-Aldrich), Necrosulfonamide (NSA) (ab143839, Abcam), SYTOX Green (S34860, ThermoFisher Scientific), SYTOX Orange (S34861, ThermoFisher Scientific), and zVAD (3188-v, Peptide Institute) were purchased from the indicated sources. The following antibodies used in this study were obtained from the indicated sources: anti-phospho-RIPK1 (31122, Cell Signaling, 1:1000), anti-RIPK1 (610459, BD Biosciences, 1:1000), anti-phospho-RIPK3 (57220, Cell Signaling, 1:1000), anti-RIPK3 (IMG-5523-2, Immugenex, 1:3000),

**Fig. 9** LCI-S uncovers two different modes of the release of HMGB1. **a–c** L929-SMART cells were transiently transfected with HMGB1-mCherry and then stimulated with TZ. The FRET/CFP ratio was analyzed as in Fig. 1b. Representative images of the FRET/CFP ratio (Ratio), intracellular (Epi), and extracellularly released (TIRF) HMGB1-mCherry (**a**) ($n = 19$ cells). Epi shows merged images of bright field and intracellular HMGB1-mCherry. Scale bar, 25 μm. Representative temporal relationship between MLKL activation and extracellular release of HMGB1-mCherry (**b**). The FRET/CFP ratio and the signals of extracellularly released HMBG1-mCherry were plotted at the indicated times after stimulation. Two vertical dotted lines at 94 and 142 min indicate initiation of an increase in the FRET/CFP ratio and extracellular release of HMGB1-mCherry, respectively. Average intervals between the initiation of the increase in the FRET/CFP ratio and extracellular release of HMGB1 (**c**) ($n = 19$ cells). **d–g** Two different modes of extracellular release of HMGB1 during necroptosis. L929-HMGB1-mCherry cells were stimulated with TZ, and extracellularly released and intracellular HMGB1-mCherry were analyzed every 2 min. Representative images of extracellularly released (TIRF) and intracellular (Epi) HMGB1-mCherry at the indicated times (**d**). Epi indicates merged images of intracellular HMGB1-mCherry and bright field. Time 0 indicates the time when the extracellular release of HMBG1-mCherry started. Scale bar, 25 μm. Representative plots of the change in intensities of extracellularly released HMGB1-mCherry in a burst-mode cell and a sustained-mode cell (**e**). The duration of the HMGB1-mCherry release in individual cell (**f**). The duration of the HMGB1-mCherry release of a single cell is plotted and clustered into two groups. Centers of each group are 109 and 7.1 min, respectively. Temporal relationship of a decrease in intensities of total and nuclear HMGB1-mCherry and an increase in extracellularly released HMGB1-mCherry (**g**). Intensities of each fraction of HMGB1-mCherry were calculated by epi-fluorescence (left vertical axis) microscopy and TIRF (right vertical axis) and plotted at the indicated times. Representative plots of a burst-mode cell (left, $n = 10$ cells) and a sustained-mode cell (right, $n = 8$ cells) in the same experiment. All results are representative of at least two independent experiments

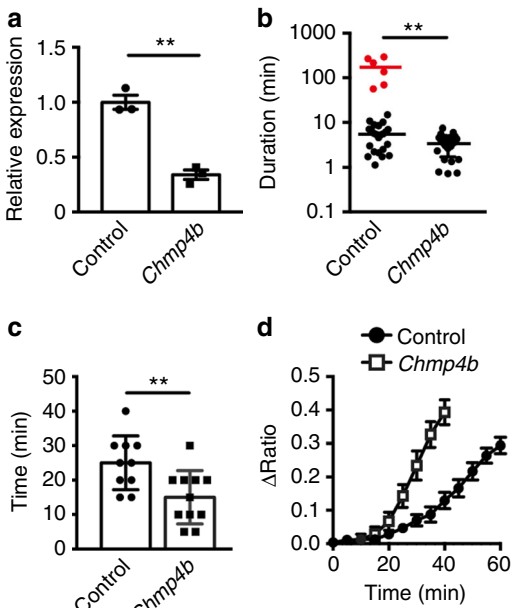

**Fig. 10** Knockdown of *Chmp4b* abrogates a sustained-mode of HMGB1 release. **a** L929-SMART/HMGB1-mCherry cells were transfected with control or *Chmp4b* siRNA, and knockdown efficiency was determined by qPCR at 24 h after transfection. Results are means ± s.d. of triplicate samples and representative of two independent experiments. Statistical significance was determined using the unpaired two-tailed Student-*t* test. \*\**P* < 0.001. **b** Duration of the HMGB1-mCherry release of a single cell. Cells were treated as in (**a**) and then stimulated with TZ. The release of HMGB1 was analyzed as in Fig. 8 and the duration of the release was determined ($n = 29$ cells for control siRNA and $n = 26$ cells for *Chmp4b* siRNA). Centers of each group of cells treated with control siRNA are 144 and 4.4 min, whereas that of *Chmp4b* siRNA is 2.9 min. Each red dot indicates individual cell showing a sutained-mode of HMGB1 release. Results are representative of two independent experiments. Statistical significance was determined using the Mann–Whitney test. \*\**P* < 0.001. **c**, **d** Cells were treated with siRNAs and stimulated with TZ. The times between the start of an increase in ΔFRET/CFP ratio and the burst of cells were calculated and plotted ($n = 10$ cells for each treatment) (**c**). Pooled results of two independent experiments. Statistical significance was determined using the Mann–Whitney test. \*\**P* < 0.001. Kinetics of averages of ΔFRET/CFP ratio of a single cell was plotted ($n = 10$ cells for control or *Chmp4b* siRNA) (**d**). Time 0 indicates the start of an increase in ΔFRET/CFP ratio. Error bars indicate s.e.m.

anti-phospho-MLKL (62233, Cell Signaling, 1:1000), anti-MLKL (3H1, made in house, 1:1000), anti-human MLKL (ab184718, Abcam, 1:1000), anti-cleaved caspase 3 (9661, Cell Signaling, 1:1000), anti-caspase-3 (9662, Cell Signaling, 1:1000), anti-actin (A2066, Sigma-Aldrich, 1:1000), anti-cytochrome c oxygenase subunit (COX) IV (Ab16056, Abcam, 1:1000), anti-tubulin (Sigma, T-5168, 1:40,000), anti-FLAG (M2, Sigma-Aldrich, 1:1000), anti-Myc (9E10, Sigma-Aldrich, 1:1000), anti-GFP (sc-8334, Santa Cruz, 1:5000), anti-GFP (66002-1-Ig, Proteintech, 1:5000), anti-mCherry (600-401-379, Rockland Immunochemicals, 1:10), anti-HMGB1 (ab18256, Abcam, 1:1000), anti-histone H3 (ab1791, Abcam, 1:1000), and HRP-conjugated donkey anti-rat IgG (712-035-153, Jackson ImmunoResearch, 1:10,000) antibodies. HRP-conjugated sheep anti-mouse IgG (NA934, 1:10,000) and HRP-conjugated donkey anti-rabbit IgG (NA934, 1:10,000) antibodies were from GE Healthcare.

**Generation of murine and human SMART**. A backbone vector for a FRET biosensor, 3536NES was kindly provided by M. Matsuda (http://www.fret.lif.kyoto-u.ac.jp/). To generate FRET biosensors for necroptosis, fragments of murine MLKL cDNAs were amplified by PCR using the following primers. α19: 5′-GGCTCGA GAGGCCGAAAGTGTTGGAATAGTG-3′ and 5′-TAGCGGCCGCACCTTCTT GTCCGTGGATTCTT-3′; α14: 5′-GGCTCGAGCAGGCCGAAAGTGTTGGAAT AGTGAGG-3′ (675F) and 5′-TAGCGGCCGTCTTTGCTGTCCGGCT-3′ (1056R); α59: 5′-GTTCCTCGAGTCAACGATATATGTCTCCCCTGAG-3′ and 5′-ACCATGCGGCCGCCCTTCTTGTCCGTGGATTCTTC-3′. A linker sequence between Ypet and ECFP of 3536NES was replaced with respective cDNA fragments. Primers used in the study were purchased from Eurofins Genomics.

Mutations of S228A, S248A, F234E, and 3ST4A were generated by introducing respective mutations into α14 by PCR-based mutagenesis using the following primers as described previously[46]. S228A: 5′-CAGGCCGAAGCTGTTGGAATAG TGAGGTTCA-3′ and 5′-CAACAGCTTCGGCCTGCTCGAGGTACA-3′; S248A: 5′-AATATTTTGCGTATATTTGGGATTTGCATTGATCA-3′ and 5′-GGGAGCA TCGAATTTCTTCATGGTTTTGAT-3′; F234E: 5′-GCGAGACTTTCAATGACG AGATCAAAACCATGAAG-3′ and 5′-GAACTATTCCAACACTTTCGGCCTG GG-3′; 3ST4A: 5′-GGGCAGCAAAGGCCACTAAAGCAGAGAGAT-3′ and 5′-G GGCGATGGCATTCTGTCCGGAACCACCAG-3′.

Amino acids 257–335 and 293–329 of α14 were replaced with 29 and 9 repeats of SAGG, respectively, resulting in the generation of Δα2α3 and Δα3 (Figure S2). Briefly, fragments corresponding to the α1 and α4 helices of murine MLKL were amplified by PCR using the following primers, respectively. α1: 675F and 5′-ATGGTACCAAATATACGCAAGATGTTGG-3′; α4: 5′-GTTGTTTCCGGATTTGAGTTAAGCAAAACA-3′ and 1056R. Amplified fragments of α1 and α4 helices were ligated into *XhoI-KpnI* and *BspEI-NotI* sites of 3536NES, respectively, in which contained oligonucleotides encoding 29 repeats of SAGG, resulting in the generation of Δα2α3. To replace α3 helix (293–329) with 9 repeats of SAGG, oligonucleotides were synthesized (Supplementary Table 1a), and ligated into the *XhoI-HindIII* sites of α14 (Δα3).

To generate Δabc, we first created *SacI* and *FspI* sites in α14 by generating mutant oligonucleotides, resulting in the generation of α14(+). We then synthesized oligonucleotides to replace amino acids spanning 243–279 (a), 287–293 (b), and 308–332 (c) with 9, 2, and 6 repeats of SAGG (Supplementary Table 1b), respectively, and ligated into the *XhoI-FspI* sites of α14(+) (Δabc). The *XhoI-SacI* and *XhoI-FspI* fragments of α14(+) were replaced with the corresponding fragments of Δabc, resulting in the generation of Δa and Δab, respectively.

To generate hSMART, the cDNA fragments of human MLKL containing α1 to α5 helices were amplified by RT-PCR from mRNA of HeLa cells as a template using the following primers (5′-GTTCCTCGAGCAGGCTGGCAGCATTGCAAT AGTGA-3′ and 5′-GTTCCTCGAGCAGGCTGGCAGCATTGCAATAGTGA-3′). Amplified fragments were ligated into 3536NES, resulting in the generation of

hα15. Then, oligonucleotides where the a and b regions corresponding to murine MLKL were replaced with SAGG repeats were synthesized and ligated into *Xho*I and *Stu*I sites of hα15, resulting in the generation of hSMART (Supplementary Table 1c and Supplementary Fig. 2c).

**Cell culture and transfection**. L929, HT29, HEK293T, and HaCaT cells were obtained from ATCC. aMoC1, murine colonic epithelial cells, were previously described[23]. Wild-type and *Mlkl*−/− MEFs were prepared from mice of the indicated genotype at E14.5 after coitus using as a standard method. MEFs were immortalized by transfection with a pEF321-T vector that encodes SV40 large T antigen (provided by S. Sugano)[47]. L929 cells were maintained with RPMI containing 10% fetal bovine serum (FBS). Wild-type and *Mlkl*−/− MEFs, HEK293T, aMoC1, HT29, and HaCaT cells were maintained with DMEM containing 10% FBS.

To generate stable transfectants of murine and human SMART, we inserted murine and human SMART cDNAs into a transposon vector, pT2KXIG (provided by K. Kawakami)[48]. We then transfected murine and human cells with pT2KXIG-mSMART and pT2KXIG-hSMART along with pCAGGS-T2TP encoding a transposase by electroporation using the Gene Pulser II (Bio-Rad), respectively. After electroporation, ECFP-positive cells were sorted by BD FACSAriaTM III (BD Biosciences).

Full lengths of histone H3 and HMGB1 were amplified by reverse transcriptase (RT)-PCR from mouse intestine and lung cDNAs using the following primers, respectively. Histone H3: 5′-GACGGTACCCATGGCTCGTACTAAGCAGA-3′ and 5′-TTAATCCCGGGCCCTCTCCCCGCGGATGCG-3′; HMGB1: 5′- TTGGTACC ATGGGCAAAGGAGATCCTAAAAAGCCG-3′ and 5′- CGACCGGTTCATCAT CATCATCTTCTTCTTCATCT-3′. These cDNAs were subcloned into a pmCherry-N1 vector (Clontech), resulting in the generation of expression vectors for histone H3-mCherry and HMGB1-mCherry. H3-mCherry and HMGB1-mCherry cDNAs were transferred to a PiggyBac transposon vector, pPB-hCMV*1-IRES-mCherry (provided by H. Niwa)[49]. L929 and L929-SMART cells were transfected with pPB-hCMV*1-histone H3-mCherry or pPB-hCMV*1-HMGB1-mCherry along with pCAGGS-PBase encoding a transposase (provided by H. Niwa)[50]. Then, mCherry-positive cells were sorted by BD FACSAria™ III (BD Biosciences).

**Knockdown by siRNAs**. L929-SMART/HMGB1-mCherry cells were transfected with control (D-001810-10-05), *Ripk3* (L-049919-00-0005), *Mlkl* (LQ-061420-00-0002, pools of J-061420-05 and -07), *Chmp4b* (L-041531-01-0020) siRNAs by lipofectamine 2000 (Invitrogen). siRNAs were purchased from Dharmacon. Knockdown of the expression of RIPK3 and MLKL was analyzed by immunoblotting with the indicated antibodies at 24 h after transfection. After transfection, cells were stimulated with TZ and subjected to LDH release assay or FRET analysis. Since anti-CHMP4B antibody did not work well in L929 cells, knockdown of the expression of *Chmp4b* was determined by quantitative polymerase chain reaction (qPCR) using the following primers. *Chmp4b*-F: 5′-GGAGAAGAGTTCGACGAG GAT-3′ and *Chmp4b*-R: 5′-TGGTAGAGGGACTGTTTCGGG-3′. qPCR analysis was performed using the 7500 Real-Time PCR detection system with SYBR green method of the target genes together with an endogenous control, murine *Hprt* with 7500 SDS software (Applied Biosystems). The amounts of *Chmp4b* were calculated relative to those of murine *Hprt* with 7500 SDS software (Applied Biosystems), and relative expression of *Chmp4b* in cells treated with *Chmp4b* siRNA vs control siRNA to be 1.0.

**Inducible expression of MLKL mutants by lentiviral vectors**. Dox-inducible lentiviral expression vectors, pF-TRE3G-PGK-puro encoding MLKL Q343A and MLKL L280P were previously described[24]. To produce lentivirus encoding MLKL mutants, we transfected HEK293T cells with pF-TRE3G-PGK-puro encoding wild-type or the indicated MLKL mutants along with packaging plasmids including pCAG-HIVgp and pCMV-VSV-G-RSV-Rev (provided by H. Miyoshi) as described previously[51]. After the infection of cells with culture supernatants containing viruses, cells were selected in the presence of 5 µg ml−1 of puromycin, resulting in the generation of *Mlkl*−/− MEFs-SMART/MLKL Q343A or MLKL L280P. To induce the expression of transfected genes, *Mlkl*−/− MEFs-SMART expressing the indicated mutants were incubated with 100 ng ml−1 of Dox for 12 h (for MLKL Q343A) or 24 h (for MLKL L280P). After confirming the expression of inducible genes, cells expressing MLKL L280P were stimulated with TBZ and subjected to the FRET analysis. Since the expression of MLKL Q343A resulted in cell death, *Mlkl* −/− MEFs-SMART/MLKL Q343A were subjected to the FRET analysis just after the addition of Dox.

**Coimmunoprecipitation and Western blotting**. HEK293T cells ($1 \times 10^6$) were plated on 60 mm dishes, and then transfected with an expression vector for each FRET probe along with an expression vector for FLAG-mRIPK3 by PEI MAX 40000 (24765, Polysciences). Cells were lysed with an IP buffer (50 mM Tris–HCl [pH 8.0], 250 mM NaCl, 0.5% Nonidet P-40, 25 mM β-glycerophosphate, 1 mM sodium orthovanadate, 1 mM sodium fluoride, 1 mM PMSF, 1 µg ml−1 aprotinin, 1 µg ml−1 leupeptin, and 1 µg ml−1 pepstatin) on ice for 30 min. After centrifugation, the supernatants were divided and incubated with the indicated antibodies for 1 h on ice, and then incubated with Protein G-Sepharose (17061801, GE

Healthcare) for another 1 h at 4 °C. The immunoprecipitates were subjected to SDS-PAGE and then transferred onto polyvinylidene difluoride membranes (IPVH 00010, Millipore). The membranes were analyzed by immunoblotting with the indicated antibodies, and developed with Super Signal West Dura Extended Duration Substrate (34076, Thermo Scientific). The signals were analyzed with Amersham Imager 600 (GE Healthcare Life Sciences).

To detect the phosphorylation of RIPK1, RIPK3, and MLKL, cells were preincubated with BV6 and zVAD for 30 min and then stimulated with TNF for the indicated times. Cells were harvested and lysed with 1× sample buffer (1% SDS, 10% sucrose, 62.5 mM Tris–HCl [pH 6.8], 2.5% β-mercaptoethanol), followed by brief sonication. Cell lysates were subjected to the SDS-PAGE. After transferring the membrane, the membrane was blocked with a blocking buffer (Cell Signaling) and then incubated with the indicated antibodies.

Uncropped images of the results of Western blotting are included in Supplementary Figures 9–13.

**Blue Native PAGE**. HT29-SMART cells were untreated or treated with TNF, BV6, and zVAD in the absence or presence of NSA for the indicated times. Cell fractionation was performed as previously described[52]. Briefly, cells were harvested with cold PBS and resuspended in a cell fractionation buffer (20 mM HEPES [pH 7.5], 100 mM KCl, 2.5 mM MgCl$_2$, and 100 mM sucrose) containing 0.025% digitonin (14592, Cayman Chemical), 25 mM β-glycerophosphate, 1 mM sodium orthovanadate, 1 mM sodium fluoride, 1 mM PMSF, 1 µg ml−1 aprotinin, 1 µg ml−1 leupeptin, and 1 µg ml−1 pepstatin on ice for 10 min. After centrifugation, the supernatant was collected as a cytosolic fraction. The resulting pellet was resuspended with the cell fractionation buffer described above and the final concentrations of digitonin were adjusted to 1% w/v and kept on ice for 20 min. After centrifugation, the 1% digitonin soluble membrane fraction and cytosolic fraction were subjected to 4–16% Bis–Tris Native PAGE gel (BN1002BOX, ThermoFisher) and then transferred onto PVDF membrane. In parallel experiments, membrane and cytosolic fractions were subjected to reducing SDS-PAGE and then transferred onto PVDF membrane. The membranes were analyzed as described in Western blotting.

**Cell death assay**. L929 cells were transiently transfected with α19, then cells were stimulated with TNF and zVAD in the presence of SYTOX Orange. Cell viability was determined by counting SYTOX-positive cells (dead cells) among CFP-positive (α19-expressing cells) or -negative (α19-nonexpressing cells). To induce RIPK3-independent necrosis of L929 cells, L929 cells were stimulated with CCCP (50 µM) in the absence or presence of zVAD (20 µM) or GSK'872 (5 µM) in Hank's balanced salt solution (HBSS) for 2 h. Cell viability was determined by WST (Water soluble Tetrazolium salts) assay (Cell Counting kit-8, Dojindo, Japan).

MEFs and aMoC1 cells were plated onto 96-well plates and cultured for 12 h in 10% DMEM. Cells were stimulated with mTNF (10 ng ml−1) in the absence or presence of BV6 (1 µM), GSK'872 (5 µM), zVAD (20 µM), or Nec-1 (20 µM) for 8 h. The values of LDH release from cells were determined by Cytotoxicity Detection Kit (Roche) as described previously[13].

HaCaT cells were stimulated with poly(I:C) (50 µg ml−1) in the absence or presence of BV6 (1 µM), zVAD (20 µM), or Nec-1 (20 µM) for 24 h. Cell viability was determined by WST assay.

**Detection of released HMGB1 and histone H3**. Cells were plated onto 24 well plates. After 12 h incubation, the culture medium was changed to Opti-MEM (Thermo Scientific), cells were stimulated with the indicated agents. Culture supernatant was collected at the indicated times after stimulation, and analyzed by immunoblotting as described above.

**Imaging analysis**. Initial experiments, L929 cells were transiently transfected with the indicated FRET biosensors with Lipofectamine 2000. Otherwise indicated, L929, MEFs, aMoC1, HT29, and HaCaT cells stably expressing mSMART or hSMART were used for imaging. Cells were seeded on gelatin-coated glass bottom dish (627870, Greinar) and then stimulated with the indicated agents. Final concentrations of agents to stimulate cells are as follows: murine TNF (10 ng ml−1), human TNF (30 ng ml−1), poly(I:C) (20 µg ml−1), zVAD (20 µM), Nec-1 (20 µM), BV6 (1 µM), GSK'872 (5 µM), and NSA (5 µM).

In the former experiments (Figs. 1, 2, Supplementary Fig. 1, 3, 6), cells were incubated in growth medium and placed in a heated chamber. Imaging analysis was carried out using a fluorescence microscope (IX-81; Olympus) with a CCD camera (ORCA-R², Hamamatsu) controlled by MetaMorph 7.0 Software (Molecular Devices). 440AF21 excitation (Ex) filter, 455DRLP dichroic mirror, and two emission (Em) filters (480AF30 for ECFP and 536AF26 for YPet) were used for imaging. The FRET emission ratio (FRET/CFP) was calculated by dividing Ex440/Em536 (FRET) by Ex440/Em480 (CFP) using MetaMorph. For the detection of SYTOX Orange uptake, an additional filter set for TRITC was used.

In the latter experiments (Figs. 3–8, 10, Supplementary Fig. 4, 5), imaging of FRET was collected using a DeltaVision microscope system (GE Healthcare) built on an Olympus IX-71 inverted microscope base equipped with Photometric Coolsnap HQ2 CCD camera, using 60×/NA1.516 PlanApo oil immersion lens (Olympus). For live cell imaging with FRET sensors, cells were seeded on

CELLview Cell Culture Dish (Greiner Bio-One) at 37 °C heat chamber with 5% $CO_2$ gas. A Blue Ex filter (400–454 nm), two Em filters (Blue Green, 463–487 nm for ECFP; Yellow Green, 537–559 nm for YPet), and C-Y-m polychroic mirror were used for imaging. The FRET emission ratio (FRET/CFP) was calculated by dividing Ex 436 nm/Em 560 nm (FRET) by Ex436 nm/Em 470 nm (CFP) using SoftWoRx (Applied Precision Inc.). For statistical analyses, the obtained images were analyzed by ImageJ and MetaMorph. ΔFRET/CFP ratio was calculated by subtracting the FRET/CFP ratio at time 0 from the FRET/CFP ratio at the indicated times. To detect Hoechst, UV filter set (Ex: 381–401 nm and Em: 409–456 nm) was used. To detect SYTOX Orange or mCherry, mCherry filter set (Ex 575 nm/Em 625 nm) was used for data collection.

**Simultaneous LCI-S, intracellular HMGB1-mCherry, and SMART**. Imaging of the release of HMGB1-mCherry by LCI-S was performed as previously described with some modifications[14]. Briefly, time-resolved measurement was performed with a completely automated inverted microscope (ECLIPSE Ti-E; Nikon, Tokyo, Japan) equipped with a high numerical aperture (NA) objective lens (CFI Apo TIRF 60× Oil, NA = 1.49, Nikon), a stage-top incubator (INUBG2TF-WSKM; Tokai Hit Co., Shizuoka, Japan) and an EM-CCD camera (ImagEM C9100-17; Hamamatsu Photonics K.K., Shizuoka, Japan). A high-pressure mercury lamp (Intensilight, Nikon) and a LED (540–600 nm, X-Cite XLED1; Excelitas technologies Corp., Waltham, MA) were used as light sources. The following sets of excitation (Ex) and emission (Em) filters and a dichroic mirror (DM) were used: For FRET, Ex: FF02-438/24-25, Em: FF01-483/32-25 (for ECFP) or FF01-542/27-25 (for Ypet), and DM: FF458-Di02-25×36; for HMGB1-mCherry, Ex: FF01-559/34-25, Em: FF01-630/69-25, and DM: FF585-Di01-25×36. These optical filters were purchased from Semrock (Rochester, NY). Eight to fifteen hours before observation, L929 cells stably expressing the indicated protein were plated to a PDMS-glass hybrid microwell array chip, on which anti-mCherry antibody was immobilized. Immediately before observation, the culture supernatant was replaced with a freshly prepared culture medium containing 1% BSA. Mineral oil was layered on top of the medium to prevent evaporation. We observed 40 and 56 microwells at 1.4 and 2 min intervals, respectively, to detect successively the signals of FRET, intranuclear and intracellular HMGB1-mCherry by epi-fluorescence microscopy and extracellular release of HMGB1-mCherry by TIRFM. The time course analysis of each fluorescence was performed using NIS Elements 4.6 (Nikon) and their relative intensities were calculated by ImageJ software. The steepness ($k$) of the HMGB1 release was estimated by fitting with the modified logistic function below by data analysis software (Origin Pro 2017, OriginLab Co., MA):

$$I(t) = I_0 e^{-\frac{t}{\tau}} / (1 + e^{-k(t-t_c)}) + I_b$$

where $I_0$ is the maximum intensity of HMGB1 release signal, $\tau$ is the time constant of the exponential decrease due to photobleaching of mCherry and release of HMGB1-mCherry from the capture antibody, $k$ is steepness of the curve and $t_c$ is the time of the sigmoid's midpoint. Then the duration $D$ needed for the signal reaching to 0.95 from 0.05 of the sigmoid curve was calculated from $k$ as follows:

$$D = \frac{2}{k} \ln\left(\frac{1}{0.05} - 1\right)$$

The logarithm of durations (logD) was classified into two groups ("burst" and "sustained") by k-means clustering methods. Average of the logD was used as the representative value of each cluster.

To determine the mode of HMGB1-mCherry release, L929-SMART/HMGB1-mCherry cells were transfected with control or *Chmp4b* siRNAs by Lipofectamine. The cells were harvested and plated into a microwell array chip at 8 h before observation. We observed 76 microwells at 2.5 min intervals to detect successively the signal of FRET and the intracellular HMGB1-mCherry by epi-fluorescence microscopy and extracellular release of HMGB1-mCherry by TIRFM. The time course analysis of each fluorescence was performed using NIS Elements 4.6 (Nikon). To determine the mode of HMGB1-mCherry release, normalized HMGB1-mCherry TIRF signals were approximated by the above modified logistic function, and the logD of HMGB1-mCherry release were compared between control and *Chmp4b* knockdown cells by Mann–Whitney test. Assembly of the logD from control and *Chmp4b* knockdown cells were then classified into two groups (burst and sustained) by k-means clustering methods. Average of the logD was used as the representative value of each cluster.

**Statistical analysis**. Statistical analysis was performed by the unpaired two-tailed Student's $t$ test, Mann–Whitney test, or the Tukey's one-way analysis of variance (ANOVA) test as appropriate. $P < 0.05$ was considered to be statistically significant.

## Data availability
The authors declare that all data supporting the findings of this study are available within the paper and its supplementary information files.

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

## Acknowledgements

We thank M. Pasparakis for *Mlkl*−/− mice; J. Murphy for pF-TRE3G-PGK-puro-WT MLKL, MLKL L280P, MLKL Q343A, and anti-MLKL antibody; H. Niwa for pPB-hCMV*1-IRES-mCherry and pCAGGS-PBase; H. Miyoshi for pCAG-HIVgp and pCMV-VSV-G-RSV-Rev; K. Kabashima for HaCaT cells; K. Kawakami for transposon-based expression vector system; and M. Matsuda for a backbone vector (3536NES) for a FRET biosensor, S. Sugano for pEF321-T, respectively. We also thank K. Ogata and S. Yamazaki, and S. Komazawa-Sakon for helpful discussion and technical assistance, respectively. This work was supported in part by Grants-in-Aid for Scientific Research (B) 17H04069 (to H.N.), Scientific Research (C) 16K01378 (to S.M.) and 16K07304 (to T. T.), Scientific Research (S) 16H06385 (to M.M.), and PRESTO JP17940748 (to Y.S.), and Challenging Exploratory Research 17K19533 (to H.N.) from Japan Society for the Promotion of Science (JSPS), Scientific Research on Innovative Areas 26110003 (to H.N.), 26110005 (to Y.Y.), 15H01366 (to Y.S.), 17H05496 (to Y.S.), 17H06017 (to T.T.), and the Japan Agency for Medical Research and Development (AMED) through AMED-CREST with grant number JP17gm0610004 (to M.M.), partly through AMED-Project for Elucidating and Controlling Mechanisms of Aging and Longevity with grant number JP17gm5010001 (to M.M.), and Private University Research Branding project (to H.N.) from the MEXT (Ministry of Education, Culture, Sports, Science and Technology), Japan, research grants from Development Research, NAKATANI Foundation (to H.N.), the Naito Science Foundation (to H.N.), the Uehara Science Foundation (to H.N.), and the Takeda Science Foundation (to H.N.). J.M.H. and J.S. are supported by Australian NHMRC grants GNT1142669, GNT1107149, and GNT1105023.

## Author contributions

S.M., Y.Y., Y.S., and H.N. designed research; S.M., Y.Y., Y.S., M.Y., R.S., J.-M.H., and R. M. performed research; J.-M.H., O.N., M.T., T.T., S.A.-A., J.S., and H.Y. contributed to new reagents/analytical tools; S.M., Y.Y., Y.S., J.-M.H., S.U., J.S., M.M., and H.N. analyzed data; S.M. and H.N. wrote the paper.

## Additional information

**Competing interests:** The authors declare no competing interests.

