## [Peer Review File · Nature Communications]

Reviewers' comments:

Reviewer #1 (Remarks to the Author):

In this report, Murai et al. established a FRET biosensor, called SMART, for necroptosis, and claim that it detects a conformational change of MLKL induced by RIP3K, and detect necroptosis. In combination with the probe for nuclear DAMPs, they show that the nuclear membranes are ruptured before the cytoplasmic membranes are ruptured. They also show that the release of HMGB1 can occur in two different ways.

General comments:

1. Phosphorylation of MLKL at T357 and S358 by RIP3K induces the conformational change of MLKL, and triggers necroptosis (References 23 and 38). In fact, the phosphomimetic MLKL (T357E/S358D) kills the cells without necroptotic stimuli, or undergo aggregation without RIP3K (Reference 30 and 38). On the other hand, the authors of this manuscript claim that SMART does not require phosphorylation in the domains of MLKL to be activated. They claim that the RIP3K's auto-phosphorylation is necessary for the activation of the probe. If so, their SMART may not be appropriate to monitor the necrotic process. Please explain in more detail. Or, show that the binding of autophosphorylated RIP3K to the MLKL mutant (T357A/T358A) activates it for necroptosis.
2. The authors claim that during necroptosis the nuclear membranes rupture before the plasma membranes rupture. This was previously reported by Yoon et al (Reference 30).
3. The authors report two different modes for the HMGB1-release during necroptosis; a burst mode and a sustained mode. It is of some interests. However, the authors did not address what causes this difference.

Minor points:

1. The authors prepared several probes. But, the first set of probes (Fig. 1) had cytotoxic activity, and was not appropriate as a probe. If so, Figure 1 and its text may not be necessary.
2. Page 5, line 3: A reference is required for the cleavage of IL33 by caspase 3.

Overall, the probe developed in this manuscript is of some interests. But, the contribution of this manuscript to the field may not be high at this stage.

Reviewer #2 (Remarks to the Author):

Murai et al. have developed SMART - a FRET-based sensor for necroptosis that employs conformational changes in the MLKL occurring upon its binding with RIP3K. The mechanism of the FRET induction appears to be dependent on this interaction, but not phosphorylation of MLKL by RIPK3. The authors also discover that nuclear membrane is ruptured during necroptosis, before the plasma membrane is ruptured. This sensor is a promising new tool for studying necroptosis using live cell imaging. The following issues need to be addressed before this manuscript is suitable for publication in Nature Communications.

Major points:

While the authors show that SMART necroptosis sensor function depends on the RIPK3 activity, the mechanism of SMART largely remains unclear. MLKL activation by RIPK3 is followed by at least two functional events (oligomerization and plasma membrane translocation) that need to be ruled in or ruled out for SMART to be a precise and useful tool for studying necroptosis. Addressing the following questions is likely to clarify the mechanism and ensure that the sensor's mechanism is indeed RIPK3-dependent and the naming of the sensor is not a misnomer (Sensor for MLKL Activation by RIPK3 based on FRET):

- 1) Does SMART depend on MLKL oligomerization? S358A mutant of MLKL is expected to not oligomerize. Does this mutant of SMART still give FRET induction upon necroptosis activation?
- 2) Does SMART depend on plasma membrane translocation of MLKL? Using NSA can be a simple tool to address this question.
- 3) Can MLKL-activating mutation Q356A induce FRET? This would rule in or rule out the necessity for MLKL-RIPK3 interaction for SMART function.
- 4) Does SMART require endogenous MLKL? Does it work in MLKL^{-/-} cells?
- 5) The study cited for F234E mutation [ref 19], which is expected to disrupt the MLKL-RIPK3 interaction, employed mouse MLKL, not human MLKL. This can explain the lack of disruption of α 14-RIPK3 binding and FRET in SMART by this mutation in the human α 14 in this study. This should be addressed and F234E shouldn't be used as a negative control for SMART.
- 6) The presumed disruptive effect of GSK'872 on the RIPK3-SMART interaction should be demonstrated by immunoprecipitation.

Minor points:

- 1) A potential dominant-negative effect of α 19 can be discussed.
- 2) Can the sensor be used in fixed cells?
- 3) The importance of using SMART for the discovery of nuclear HMGB release is unclear.

Reviewer #3 (Remarks to the Author):

This paper develops fluorescence resonance energy transfer (FRET) biosensors that monitor necroptosis based on the MLKL activation by RIPK3. The biosensors were applied to verify that MLKL activation can represent the necroptosis but not apoptosis or necrosis. The authors further identified two modes of High Mobility Group Box 1 release in necroptosis cells. While the biosensors are useful and observations generally interesting, the overall study lacks a mechanistic insight on the role of MLKL in regulating necroptosis and on the two modes of HMGB1 release. The manuscript also lacks a coherent organization to present a concrete story on the molecular mechanisms underlying MLKL activation or necroptosis. Some specific items are listed as following:

1. Across the whole manuscript, there is a lack of clear and established specific molecular marker(s) for necroptosis, in addition to MLKL activation. The leakage of macromolecules via nuclear or plasma membrane demonstrated in the manuscript should be common

features to necroptosis and other cell death processes, e.g. necrosis. It will be nice to apply these established and specific necroptosis markers to the same cell together with the MLKL FRET biosensor(s) so that a clear spatiotemporal correlation can be established between necroptosis marker(s) and FRET signals to examine how MLKL FRET signals can specifically report necroptosis processes.

2. It will be nice if inhibitors or siRNA/CRISPR can be applied to manipulate RIPK3/MLKL pathway and examine necroptosis phenotypes, such as the specific marker signals and HMGB1 leakage. This should allow the examination or establishment of the causative relationship between RIPK3/MLKL and necroptosis.

3. The reporting mechanism of the biosensor(s) is not clear. The authors established that the biosensor function is not dependent on phosphorylation by RIK3 but would be nice to know how the biosensors work so that other people can use them with clear understanding.

4. In figure 3, why Delta alpha 3, still interacting with RIK3, but has no response. It is nice to provide more molecular insights which can help to understand how the biosensor works.

5. Is MLKL activation causatively related to the two modes of HMGB1? If so, how the MLKL activation characteristics determines these two different modes? More work will be needed to have a better understanding on how these two modes of HMGB1 release are formed, e.g. what are the contributors and factors involved and how they contribute to the determination of these different modes.

Reviewers' comments:

Reviewer #1 (Remarks to the Author):

In this report, Murai et al. established a FRET biosensor, called SMART, for necroptosis, and claim that it detects a conformational change of MLKL induced by RIP3K, and detect necroptosis. In combination with the probe for nuclear DAMPs, they show that the nuclear membranes are ruptured before the cytoplasmic membranes are ruptured. They also show that the release of HMGB1 can occur in two different ways.

General comments:

COMMENT:

- 1. Phosphorylation of MLKL at T357 and S358 by RIP3K induces the conformational change of MLKL, and triggers necroptosis (References 23 and 38). In fact, the phosphomimetic MLKL (T357E/S358D) kills the cells without necroptotic stimuli, or undergo aggregation without RIP3K (Reference 30 and 38). On the other hand, the authors of this manuscript claim that SMART does not require phosphorylation in the domains of MLKL to be activated. They claim that the RIPK3's auto-phosphorylation is necessary for the activation of the probe. If so, their SMART may not be appropriate to monitor the necrotic process. Please explain in more detail. Or, show that the binding of autophosphorylated RIP3K to the MLKL mutant (T357A/S358A) activates it for necroptosis.**

RESPONSE; Thank you for pointing out a critical issue of our study. Indeed, phosphorylation T357 and S358 of human MLKL (a counterpart of S345 of murine MLKL) is essential for execution of necroptosis. In sharp contrast, mutation of phosphorylation sites of the SMART probe itself did not abrogate TNF/zVAD-induced SMART activation. To address apparently inconsistent results, we performed additional experiments to elucidate the mechanisms underlying SMART activation and obtained the following results.

- 1) A RIPK3 inhibitor, GSK'872 blocked SMART activation in L929 cells (Fig. 1j).
- 2) Knockdown of *Ripk3* or *Mkl1* by siRNAs blocked TNF/zVAD -induced SMART activation in L929 cells (Fig. 4a-c).
- 3) TNF/BV6/zVAD-induced SMART activation was blocked in *Mkl1*^{-/-} MEFs (Fig. 4d, e).
- 4) Activation of SMART was correlated with phosphorylation of RIPK1, RIPK3, and MLKL, hallmarks of necroptosis in L929 cells and MEFs (Fig. 4f-i).
- 5) Inducible expression of a constitutive active mutant of MLKL Q343A induced SMART activation even in the absence of TNF stimulation in *Mkl1*^{-/-} MEFs (Fig. 5).
- 6) Necrosulfonamide (NSA), an inhibitor for MLKL, binds to N-terminal cysteine at 86 (C86) of human MLKL, but not mMLKL, and inhibits necroptosis in human cells. To test the effect of NSA on SMART activation, we generated human version of SMART. Although human SMART does not possess C86, NSA blocked TNF/BV6/zVAD -induced hSMART activation in HT29 cells (Fig. 6a-e).

These results suggest that activation of SMART requires RIPK3 kinase activity and phosphorylation of MLKL, whereas phosphorylation of the SMART probe itself is not required for SMART activation. Moreover, oligomers of MLKL is sufficient for SMART activation. Thus, we surmise that SMART senses oligomers of MLKL even in the absence of RIPK3 activation, resulting in conformational changes, and FRET induction (see our posed model in Supplementary Fig. 6). We have included the results in the indicated Figures and mentioned it in the text (lines 210-251 and 366-376).

- 2. COMMENT; The authors claim that during necroptosis the nuclear membranes rupture before the plasma membranes rupture. This was previously reported by Yoon et al (Reference 30).**

RESPONSE; We added the reference in the text (lines 392-395).

3. COMMENT; The authors report two different modes for the HMGB1-release during necroptosis; a burst mode and a sustained mode. It is of some interests. However, the authors did not address what causes this difference.

RESPONSE; Thank you for pointing out a critical issue. To compare cells showing a sustained-mode and a burst-mode quantitatively, we first estimated the duration of HMGB1-mCherry release in individual cells and classified them into two groups by k-means clustering. The representative duration of extracellular HMGB1 release of the burst-mode and the sustained-mode are 7.1 and 109 min, respectively (Fig. 8f, black dots vs red dots).

Regarding the molecular mechanism underlying two modes of the release of HMGB1, we focused on the endosomal sorting complex required for transport (ESCRT). The ESCRT complex is involved in mediating receptor sorting, membrane remodeling, and membrane scission (Christ et al., 2017). A recent study has shown that components of the ESCRT-III complex are involved in extrusion of MLKL-containing vesicles, thereby attenuating TNF-induced necroptosis (Gong et al., 2017). Taken that knockdown of the ESCRT-III proteins, such as *Chmp4b* delays TNF-induced necroptosis (Gong et al., 2017), we surmised that the ESCRT-III proteins might be involved in determining whether cells show a sustained-mode or burst-mode release of HMGB1. Notably, knockdown of *Chmp4b* abrogated cells showing a sustained-mode release of HMGB1 (Fig. 9b, red dots). Thus, the balance between the extent of pore forming activity of MLKL and membrane repairing capacity of proteins of the ESCRT complex, such as CHMP4B, might determine whether cells release HMGB1 via the sustained-mode or burst-mode. It would be intriguing to examine mRNA expression of *Mlkl* and *Chmp4b* in a single cell that releases HMGB1 in a burst-mode or sustained-mode. However, such analysis is beyond the scope of the present study and should be performed in the future study. We included the results (Fig. 9 and Supplementary Fig. S9) and mentioned it in the text (lines 303-344 and 396-405).

Minor points:

1. **COMMENT; The authors prepared several probes. But, the first set of probes (Fig. 1) had cytotoxic activity, and was not appropriate as a probe. If so, Figure 1 and its text may not be necessary.**

RESPONSE; As suggested, we moved some results of Fig. 1 to Supplementary Figure 1.

2. **COMMENT; Page 5, line 3: A reference is required for the cleavage of IL33 by caspase 3.**

RESPONSE; As suggested, we added a reference (lines 71-73).

Reviewer #2 (Remarks to the Author):

Murai et al. have developed SMART - a FRET-based sensor for necroptosis that employs conformational changes in the MLKL occurring upon its binding with RIPK3. The mechanism of the FRET induction appears to be dependent on this interaction, but not phosphorylation of MLKL by RIPK3. The authors also discover that nuclear membrane is ruptured during necroptosis, before the plasma membrane is ruptured. This sensor is a promising new tool for studying necroptosis using live cell imaging. The following issues need to be addressed before this manuscript is suitable for publication in Nature Communications.

Major points:

While the authors show that SMART necroptosis sensor function depends on the RIPK3 activity, the mechanism of SMART largely remains unclear. MLKL activation by RIPK3 is followed by at least two functional events (oligomerization and plasma membrane translocation) that need to be ruled in or ruled out for SMART to be a precise and useful tool for studying necroptosis. Addressing the following questions is likely to clarify the mechanism and ensure that the

sensor's mechanism is indeed RIPK3-dependent and the naming of the sensor is not a misnomer (Sensor for MLKL Activation by RIPK3 based on FRET):

1) COMMENT; Does SMART depend on MLKL oligomerization? S358A mutant of MLKL is expected to not oligomerize. Does this mutant of SMART still give FRET induction upon necroptosis activation?

RESPONSE; Thank you for your valuable comments to encourage us to revise the manuscript. Our preliminary results showed that a mutant of murine MLKL S345A (a murine counterpart of human MLKL T357A and S358A) did not act as a dominant negative mutant (unpublished results by Hildebrand and Silke). Taken that MLKL L280P acts as a dominant negative mutant (Murphy et al., 2013), we expressed MLKL L280P in *Mkl1*^{-/-} MEFs in a Doxycyclin (Dox)-dependent manner. We found that TNF/BV6/zVAD stimulation did not either SMART activation or necroptosis after Dox induction. However, mutations of all phosphorylation sites of the α 14 and SMART probe did not impair TNF/zVAD-induced SMART activation (Fig. 1f, g, and data not shown). In sharp contrast, expression of a constitutive active mutant, MLKL Q343A that spontaneously forms oligomers, did induce SMART activation even in the absence of TNF stimulation. Thus, we conclude that oligomers of endogenous MLKL, but not phosphorylation of the SMART probe itself, is required and sufficient for SMART activation. We made a new Figure 5 to include these results and proposed a model for SMART activation in Supplementary Fig. 6 and mentioned it in the text (lines 222-237 and 372-376).

2) Does SMART depend on plasma membrane translocation of MLKL? Using NSA can be a simple tool to address this question.

RESPONSE; Thank you for suggesting a critical experiment. NSA targets cysteine at 86 (C86) of human MLKL, thereby blocking oligomers and subsequent membrane translocation of MLKL (Liu et al., 2017; Sun et al.,

2012). Since human MLKL, but not murine MLKL, possesses C86, we generated a human version of SMART (hSMART) and transfected a human cell line, HT29 cells with hSMART. As expected, NSA blocked TNF/BV6/zVAD-induced necroptosis of HT29 cells. Surprisingly, NSA blocked TNF/BV6/zVAD-induced SMART activation, even though hSMART does not possess C86 (Fig. 6a to e). Taken that NSA blocked oligomers of hMLKL, these results further substantiate that oligomers of MLKL is required for SMART activation.

Since the results in old Fig. 3 were obtained by transient transfection experiments of human α 14, we replaced them with a new Fig. 6 that includes the results using HT29 and HaCaT cells stably expressing hSMART and the effect of NSA on TNF/BV6/zVAD-induced SMART activation (Fig. 6f-h). We mentioned it in the text (lines 239-251).

3) Can MLKL-activating mutation Q356A induce FRET? This would rule in or rule out the necessity for MLKL-RIPK3 interaction for SMART function.

RESPONSE; Please see the response to the comment 2)

4) Does SMART require endogenous MLKL? Does it work in MLKL^{-/-} cells?

RESPONSE; Thank you for indicating a critical point. As suggested, we generated *Mlkl*^{-/-} MEFs stably expressing SMART. We stimulated *Mlkl*^{-/-} MEFs-SMART with TNF/BV6/zVAD and found that activation of SMART was abolished in *Mlkl*^{-/-} MEFs. In addition, we knocked down *Mlkl* by siRNA and obtained similar results. Thus, as mentioned as above, endogenous MLKL, is required for SMART activation. We made a new Fig. 4 to include these results and proposed a model for the mechanism underlying SMART

activation in Supplementary Fig. 6. We mentioned these results in the text (lines 210-220 and 368-371).

- 5) The study cited for F234E mutation [ref 19], which is expected to disrupt the MLKL-RIPK3 interaction, employed mouse MLKL, not human MLKL. This can explain the lack of disruption of α 14-RIPK3 binding and FRET in SMART by this mutation in the human α 14 in this study. This should be addressed and F234E shouldn't be used as a negative control for SMART.**

RESPONSE; Sorry for causing confusion. As mentioned in the text, F234E mutant was generated using murine MLKL, but not human MLKL. In contrast to previous study (Xie et al., 2013), MLKL F234E interacted with RIPK3 at least under our experimental conditions. Taken that FRET was still induced in L929 cells expressing MLKL F234E after TNF/zVAD stimulation (Fig. 2), these results further substantiate that binding of RIPK3 to the FRET biosensors is required for FRET induction.

- 6) The presumed disruptive effect of GSK'872 on the RIPK3-SMART interaction should be demonstrated by immunoprecipitation.**

RESPONSE; As suggested, we transfected HEK293T cells with expression vectors for RIPK3 and SMART, and then treated cells with GSK'872 for 4 hours. Unexpectedly, binding of RIPK3 with SMART was not inhibited in HEK293T cells under the conditions when these expression vectors were overexpressed (Fig. 2d). This indicates that kinase activity of RIPK3 and phosphorylation of the SMART probe are not required for their binding. However, TNF/zVAD -induced SMART activation was abolished in L929 cells in the presence of GSK'872 (Fig. 1j). GSK'872 has been shown to block phosphorylation and subsequent oligomers of MLKL. Thus, GSK'872 inhibits SMART activation through suppressing oligomers of MLKL, but not interaction of RIPK3 with SMART in

HEK293T cells. We included the results in Fig. 2d and mentioned it in the text (lines 176-181).

Minor points:

1) A potential dominant-negative effect of $\alpha 19$ can be discussed.

RESPONSE; We had surmised that $\alpha 19$ inhibited the interaction of RIPK3 with MLKL. We found that MLKL interacted with $\alpha 19$, $\alpha 14$, and $\alpha 59$ (Fig. 1d). Moreover, RIPK3 was efficiently immunoprecipitated with MLKL even in the presence of $\alpha 19$ (Fig. 1e). Taken that oligomers of MLKL are sufficient for necroptosis induction, we hypothesize that $\alpha 19$ might block oligomers of MLKL, but not the interaction of RIPK3 with MLKL. We included the results in Fig. 1e and mentioned it in the text (lines 141-143).

2) Can the sensor be used in fixed cells?

RESPONSE; As suggested, we performed FRET analysis using L929-SMART cells that had been unstimulated or stimulated with TNF/zVAD for 2 hours, and then fixed with paraformaldehyde (PFA). As shown in Rebuttal Fig. 1, an increase in the FRET/CFP ration was not detected. Thus, we surmised that SMART does not appear to work in fixed cells. We only include the results in a rebuttal letter.

3) The importance of using SMART for the discovery of nuclear HMGB release is unclear.

RESPONSE; Indeed, SMART is not required for detection of release of nuclear HMGB1. However, SMART is suitable for evaluating the kinetics of execution of necroptosis including oligomers of MLKL and release of nuclear HMGB1 (Fig. 8a, and 8b). We mentioned it in the text (lines 294-297).

Reviewer #3 (Remarks to the Author):

This paper develops fluorescence resonance energy transfer (FRET) biosensors that monitor necroptosis based on the MLKL activation by RIPK3. The biosensors were applied to verify that MLKL activation can represent the necroptosis but not apoptosis or necrosis. The authors further identified two modes of High Mobility Group Box 1 release in necroptosis cells. While the biosensors are useful and observations generally interesting, the overall study lacks a mechanistic insight on the role of MLKL in regulating necroptosis and on the two modes of HMGB1 release. The manuscript also lacks a coherent organization to present a concrete story on the molecular mechanisms underlying MLKL activation or necroptosis. Some specific items are listed as following:

RESPONSE; Thank you for pointing out a critical issue of our study. Indeed, phosphorylation T357 and S358 of human MLKL (a counterpart of S345 of murine MLKL) is essential for execution of necroptosis. In sharp contrast, mutation of phosphorylation sites of the SMART probe itself did not abrogate TNF/zVAD-induced SMART activation. To address apparently inconsistent results, we performed additional experiments to elucidate the mechanisms underlying SMART activation and obtained the following results.

- 1) A RIPK3 inhibitor, GSK'872 blocked SMART activation in L929 cells (Fig. 1j).

- 2) Knockdown of *Ripk3* or *Mlkl* by siRNAs blocked TNF/zVAD -induced SMART activation in L929 cells (Fig. 4a-c).
- 3) TNF/BV6/zVAD-induced SMART activation was blocked in *Mlkl*^{-/-} MEFs (Fig. 4d, e).
- 4) Activation of SMART was correlated with phosphorylation of RIPK1, RIPK3, and MLKL, hallmarks of necroptosis in L929 cells and MEFs (Fig. 4f-i).
- 5) Inducible expression of a constitutive active mutant of MLKL Q343A induced SMART activation even in the absence of TNF stimulation in *Mlkl*^{-/-} MEFs (Fig. 5).
- 6) Necrosulfonamide (NSA), an inhibitor for MLKL, binds to N-terminal cysteine at 86 (C86) of human MLKL, but not mMLKL, and inhibits necroptosis in human cells. To test the effect of NSA on SMART activation, we generated human version of SMART. Although human SMART does not possess C86, NSA blocked TNF/BV6/zVAD -induced hSMART activation in HT29 cells (Fig. 6a-e).

These results suggest that activation of SMART requires RIPK3 kinase activity and phosphorylation of MLKL, whereas phosphorylation of the SMART probe itself is not required for SMART activation. Moreover, expression of oligomers of MLKL is sufficient for SMART activation. Thus, we surmise that SMART senses oligomers of MLKL even in the absence of RIPK3 activation, resulting in conformational changes and FRET induction (see our posed model in Supplementary Fig. 6). We have included the results in the indicated Figures and mentioned it in the text (lines 210-251 and 366-376).

COMMENT;

1. **COMMENT. Across the whole manuscript, there is a lack of clear and established specific molecular marker(s) for necroptosis, in addition to MLKL activation. The leakage of macromolecules via nuclear or plasma membrane demonstrated in the manuscript should be common features to necroptosis and other cell death processes, e.g. necrosis. It**

will be nice to apply these established and specific necroptosis markers to the same cell together with the MLKL FRET biosensor(s) so that a clear spatiotemporal correlation can be established between necroptosis marker(s) and FRET signals to examine how MLKL FRET signals can specifically report necroptosis processes.

RESPONSE; As suggested, we investigated the kinetics of phosphorylation of RIPK1, RIPK3, and MLKL in L929 cells and MEFs after TNF/BV6/zVAD stimulation. While phosphorylation of RIPK1 was detected at 0.5 ~ 1 hr, phosphorylation of RIPK3 and MLKL was detected at 1.0 ~ 1.5 hr in TNF/BV6/zVAD -stimulated L929 cells. On the other hand, TNF/BV6/zVAD stimulation induced phosphorylation of RIPK1, RIPK3, and MLKL at 1 hr and sustained thereafter in MEFs. Thus, kinetics of phosphorylation of RIPK3 and MLKL was slightly earlier or comparable to the kinetics of activation of SMART, which is consistent with the idea that SMART senses oligomers of MLKL. We included these results in Fig. 4f-i, and mentioned it in the text (lines 217-220).

2. COMMENT; It will be nice if inhibitors or siRNA/CRISPR can be applied to manipulate RIPK3/MLKL pathway and examine necroptosis phenotypes, such as the specific marker signals and HMGB1 leakage. This should allow the examination or establishment of the causative relationship between RIPK3/MLKL and necroptosis.

RESPONSE; As suggested, we knocked down expression of *Ripk3* and *Mlkl* by siRNAs in L929-SMART cells. As expected, knockdown of RIPK3 abolished TNF/zVAD-induced necroptosis and SMART activation, which is consistent with our hypothesis. To our surprise, knockdown of MLKL similarly abolished TNF/zVAD-induced SMART activation in L929 cells. We also confirmed that TNF/BV6/zVAD stimulation did not activate SMART in *Mlkl*^{-/-} MEFs. These results further substantiate that endogenous RIPK3 and MLKL are required for SMART activation. Moreover, taken that a RIPK3 inhibitor, GSK'872 suppressed TNF/zVAD-induced SMART activation, phosphorylation of endogenous MLKL, but not the SMART probe itself is required for activation of SMART. We included

these results in Fig.1j, Fig. 4a-e, and proposed a model for SMART activation in Supplementary Fig. 6, and mentioned it in the text (lines 210-217 and 368-371).

3. COMMENT; The reporting mechanism of the biosensor(s) is not clear. The authors established that the biosensor function is not dependent on phosphorylation by RIK3 but would be nice to know how the biosensors work so that other people can use them with clear understanding.

RESPONSE; See the responses to general comments.

COMMENT; 4. In figure 3, why Delta alpha 3, still interacting with RIK3, but has no response. It is nice to provide more molecular insights which can help to understand how the biosensor works.

RESPONSE; Sorry for causing confusion. The FRET/CFP ratio of cells expressing $\Delta\alpha 3$ did increase, although an increase in the FRET/CFP ratio was low compared to those of $\alpha 14$ (please see old Fig. 2d and new Fig. 2b). As mentioned in the text (lines 168-175), interaction of a FRET biosensor with RIPK3 might be prerequisite, but not sufficient for FRET activation. Conformation of $\Delta\alpha 3$ might be less favorable for FRET induction compared to $\alpha 14$.

As described in the response to the comments, we propose a tentative model for SMART activation in Supplementary Fig. 6 and mentioned in the text (see the responses to general comments).

COMMENT; 5. Is MLKL activation causatively related to the two modes of HMGB1? If so, how the MLKL activation characteristics determines these two different modes? More work will be needed to have a better understanding on how these two modes of HMGB1 release are formed, e.g. what are the contributors and factors involved and how they contribute to the determination of these different modes.

RESPONSE; Thank you for pointing out a critical issue. To compare cells showing a sustained-mode and a burst-mode quantitatively, we first estimated the duration of HMGB1-mCherry release in individual cells and classified them into two groups by k-means clustering. The representative duration of extracellular HMGB1 release of the burst-mode and the sustained-mode are 7.1 and 109 min, respectively (Fig. 8f, black dots vs red dots).

Regarding the molecular mechanism underlying two modes of the release of HMGB1, we focused on the endosomal sorting complex required for transport (ESCRT). The ESCRT complex is involved in mediating receptor sorting, membrane remodeling, and membrane scission (Christ et al., 2017). A recent study has shown that components of the ESCRT-III complex are involved in extrusion of MLKL-containing vesicles, thereby attenuating TNF-induced necroptosis (Gong et al., 2017). Taken that knockdown of the ESCRT-III proteins, such as *Chmp4b* delays TNF-induced necroptosis (Gong et al., 2017), we surmised that the ESCRT-III proteins might be involved in determining whether cells show a sustained-mode or burst-mode release of HMGB1. Notably, knockdown of *Chmp4b* abrogated cells showing a sustained-mode release of HMGB1 (Fig. 9b, red dots). Thus, the balance between the extent of pore forming activity of MLKL and membrane repairing capacity of proteins of the ESCRT complex, such as CHMP4B, might determine whether cells release HMGB1 via the sustained-mode or burst-mode. It would be intriguing to examine mRNA expression of *Mlkl* and *Chmp4b* in a single cell that releases HMGB1 in a burst-mode or sustained-mode. However, such analysis is beyond the scope of the present study and should be performed in the future study. We included the results (Fig. 9 and Supplementary Fig. S9) and mentioned it in the text (lines 303-344 and 396-405).

References

- Christ, L., Raiborg, C., Wenzel, E.M., Campsteijn, C., and Stenmark, H. (2017). Cellular Functions and Molecular Mechanisms of the ESCRT Membrane-Scission Machinery. *Trends Biochem Sci* 42, 42-56.
- Gong, Y.N., Guy, C., Olauson, H., Becker, J.U., Yang, M., Fitzgerald, P., Linkermann, A., and Green, D.R. (2017). ESCRT-III Acts Downstream of

MLKL to Regulate Necroptotic Cell Death and Its Consequences. *Cell* *169*, 286-300 e216.

Liu, S., Liu, H., Johnston, A., Hanna-Addams, S., Reynoso, E., Xiang, Y., and Wang, Z. (2017). MLKL forms disulfide bond-dependent amyloid-like polymers to induce necroptosis. *Proc Natl Acad Sci U S A* *114*, E7450-E7459.

Murphy, J.M., Czabotar, P.E., Hildebrand, J.M., Lucet, I.S., Zhang, J.G., Alvarez-Diaz, S., Lewis, R., Lalaoui, N., Metcalf, D., Webb, A.I., *et al.* (2013). The pseudokinase MLKL mediates necroptosis via a molecular switch mechanism. *Immunity* *39*, 443-453.

Sun, L., Wang, H., Wang, Z., He, S., Chen, S., Liao, D., Wang, L., Yan, J., Liu, W., Lei, X., *et al.* (2012). Mixed lineage kinase domain-like protein mediates necrosis signaling downstream of RIP3 kinase. *Cell* *148*, 213-227.

Xie, T., Peng, W., Yan, C., Wu, J., Gong, X., and Shi, Y. (2013). Structural insights into RIP3-mediated necroptotic signaling. *Cell Rep* *5*, 70-78.

Correspondence Table between New Figures and Old Figures.

New Figures	Old Figures	New Movies	Old Movies
Fig. 1a-c	Fig. 1a, 1d, 1b	Movie S1	Movie S3
Fig. 1d, e	new results	Movie S2	Movie S4
Fig 1f, h, g	Fig. S3a, S3b, S3d	Movie S3	Movie S5
Fig. 1i	Fig. 1e	Movie S4	Movie S6
Fig. 1j	Fig. 2e	Movie S5	Movie S7
		Movie S6	Movie S8
Fig. 2a-c	Fig. 2a, 2d, 2b; Fig. 4a, 4d, 4b	Movie S7	Movie S9
Fig. 2d	new results	Movie S8	Movie S10
Fig. 3a-g	Fig. 5a-g	Movie S9 replaced with more representative one	Movie S11
Fig. 4a-i	new results	Movie S10 replaced with more representative one	Movie S12
Fig. 5a-e	new results		
Fig. 6a-e, i-h	new results		
Fig. 6f	Fig. 3f		
Fig. 7a-d	Fig. 6a-d		
Fig. 8a-c	Fig. 7a-c		
Fig. 8d	replaced with more representative ones		
Fig. 8e, 8g	Fig. 7e, 7g		
Fig. 8f	new results		
Fig. 9a-d	new results		
Fig. S1a, b	Fig. S1a, b		
Fig. S1c, d	Fig. 1c, Fig. S3c		
Fig. S2a, b	Fig. S2		
Fig. S2c	new result		
Fig. S3a, b	Fig. 2c, Fig. 4c		
Fig. S4a-e	Fig. S4a-e		
Fig. S5	new results		
Fig. S6	new Figure		
Fig. S7a-c	Fig. 3a, 3c, 3d		
Fig. S8	Fig. S5		
Fig. S9	new results		
Fig. S10	uncropped images of WB		
Fig. S11	uncropped images of WB		
Fig. S12	uncropped images of WB		
Fig. S13	uncropped images of WB		

Reviewers' comments:

Reviewer #1 (Remarks to the Author):

The authors responded well to my concern about the phosphorylation of MLKL (Comment 1). I agree that the probe senses the oligomerized MLKL. But how? Do the author think that the probe is incorporated into the MLKL oligomer? Please discuss.

To the Comment 3, the authors discuss that the membrane repair system may be responsible for the sustained and burst mode of the rupture. This is interesting.

Other minor points:

Line 139: three mutants interacted----. This sentence is not clear.

Line 141: How Fig.1e indicates that a19 did not block the interaction between MLKL and RIP3K.

Line 149: The sentences are not clear.

Line 173; From the data in Fig. 2c, can the authors suggest that "the interaction of (with?) RIPK3 is not sufficient for FRET induction?"

Line 176-181: I understand that the kinase activity of RIP3 is not necessary for the interaction with SMART with RIPK3, but is required for the SMART activation. Please rewrite the paragraph from Line 159 more concisely to clarify the point.

Line 427: I believe that Histone H3 is released from apoptotic cells as a component of nucleosomes in apoptotic bodies. The comparison here may not be appropriate.

Reviewer #2 (Remarks to the Author):

The authors have addressed most of my points.

The following points still need to be addressed, in the light of introduction of a new model for SMART activation:

1) Since the model of SMART activation has changed from being dependent on RIPK3-MLKL binding to MLKL oligomerization, this should be reflected in the wording in the abstract. The statement "Activation of SMART requires endogenous RIPK3 and MLKL, and depends on oligomers of MLKL." is not correct, since Q343A mutant can also activate the SMART probe in the absence of RIPK3. In other words, mechanistically, SMART activation depends on oligomerization of MLKL, which is induced by RIPK3 or other means, such as mutational activation.

2) The phrase "and inhibits necroptosis through preventing oligomers of MLKL in human cells" (line 248) is not correct. NSA inhibits MLKL function by preventing its plasma membrane localization. Also, the ref 25 at the end of that sentence is not correctly placed. NSA was not shown to inhibit MLKL oligomerization in that study. Which study shows that NSA inhibits MLKL oligomerization?

3) The authors should provide evidence that NSA inhibited oligomerization of MLKL in the presence of the hSMART probe to prove that hSMART works by detecting MLKL oligomerization. A simple way to do this is to employ the non-reducing PAGE system used by Wang et al., 2014, Mol Cell study. NSA does not seem to block oligomerization of MLKL

in that study.

5) NSA blocking SMART activation suggests that MLKL plasma membrane translocation is required for the activation of the sensor.

4) Does T357E/S358D-induced MLKL oligomerization and hSMART activation get blocked by NSA?

5) The key experiment to support the new model is to do non-reducing PAGE after TSZ ± NSA and immunoblot for the SMART probe (e.g. CFP), showing that the probe is associated with the endogenous MLKL oligomers and that NSA is blocking this association.

Alternatively, a size-exclusion chromatography experiment will be needed to support the model. Currently, the only evidence that supports the new model is the Q343A experiment and this experiment lacks evidence that the oligomerization, but not the plasma membrane translocation of MLKL is required for SMART activation. The HT-29 ± NSA experiment does not prove the oligomerization model, since NSA blocks plasma membrane translocation.

Reviewer #3 (Remarks to the Author):

The authors have sufficiently addressed my concerns.

Reviewers' comments:

Reviewer #1 (Remarks to the Author):

COMMENT: The authors responded well to my concern about the phosphorylation of MLKL (Comment 1). I agree that the probe senses the oligomerized MLKL. But how? Do the author think that the probe is incorporated into the MLKL oligomer? Please discuss.

RESPONSE: Thank you for your appreciation of the revised experiments. As pointed out by Reviewer #2, we further investigated the molecular mechanisms how SMART monitored the conformational changes of MLKL such as oligomerization. We treated HT29-SMART cells with TBZ in the absence or presence of NSA, and cytosolic and membrane fractions were subjected to Blue-Native PAGE. As expected, MLKL formed oligomers on the plasma membrane of cells at 6 hours following TBZ stimulation (Supplementary Figure 7a). In sharp contrast, NSA blocked plasma membrane translocation of oligomerized MLKL and an increase in the FRET/CFP ratio of SMART. Thus, SMART monitors plasma membrane translocation of oligomerized MLKL, but not oligomerized MLKL itself. Since anti-GFP antibody did not work well on BN-PAGE, we could detect SMART by anti-GFP antibody in the cytoplasmic fraction on SDS-PAGE under reducing conditions (Figure 7b). This suggests that SMART did not translocate into the plasma membrane via direct interaction with oligomerized MLKL. This result was also consistent with that FRET signals of SMART were diffusely distributed in the cytosol of cells undergoing necroptosis (Figure 3a). One of the plausible explanations would be that SMART might monitor drastic changes of cellular conditions induced by plasma membrane translocation of oligomerized MLKL, such as relative ratios of RIPK3 and MLKL in the cytosol. Further study will be required to address this issue. We made a new Supplementary Figure 7 to include the results and mentioned it in the text (Lines 249-258 and 389-396).

MuraiFigS7

Supplementary Figure 7. NSA blocks plasma membrane translocation of oligomerized MLKL. HT29-SMART cells were stimulated with TBZ in the absence or presence of NSA for the indicated times. Then cytosolic and membrane fractions were prepared and subjected to BN-PAGE (**a**) or SDS-PAGE under reducing conditions (**b**). Expression of each protein in cytosolic (C) or membrane (M) fractions was determined by immunoblotting with the indicated antibodies. Oligo and mono indicate oligomers and monomer of MLKL, respectively. Results are representative of two independent experiments.

COMMENT: To the Comment 3, the authors discuss that the membrane repair system may be responsible for the sustained and burst mode of the rupture. This is interesting.

RESPONSE: Thank you for your appreciation of our findings.

Other minor points:

COMMENT: Line 139: three mutants interacted----. This sentence is not clear.

RESPONSE: Thank you for picking this up, we changed the sentence to " Cotransfection experiments revealed that three biosensors interacted with RIPK3, and MLKL in the absence or presence of RIPK3 and without induction of necroptosis (Fig. 1c-e)..." (Lines 139-141).

COMMENT: Line 141: How Fig.1e indicates that α 19 did not block the interaction between MLKL and RIP3K.

RESPONSE: Sorry for the ambiguous expression. We added the following sentence to clarify this point. "Notably, Flag-RIPK3 was efficiently coimmunoprecipitated with Myc-MLKL in the presence of α 19 (Fig. 1e), suggesting that α 19 did not block the binding of MLKL to RIPK3. Thus, we surmised that α 19 might block oligomerization of endogenous MLKL, thereby inhibiting TZ-induced necroptosis (Lines 142-146).

COMMENT: Line 149: The sentences are not clear.

RESPONSE: Sorry for ambiguous expression. We deleted one sentence, and have now changed the sentences to "Thus, while phosphorylation of MLKL is required for oligomerization of MLKL, phosphorylation of α 14 is not necessary for FRET induction. Nevertheless, the RIPK1 inhibitor, Necrostatin-1 (Nec-1) or a RIPK3 inhibitor, GSK'872 blocked necroptosis and abolished an increase in the FRET/CFP ratio in cells expressing α 14 upon TZ stimulation (Fig. 1i, j). Taken together these data paradoxically indicate that phosphorylation of α 14 is not required for FRET activity yet kinase activities of RIPK1 and RIPK3 are nevertheless required for α 14 FRET activation. " (Lines 151-157)

COMMENT: Line 173; From the data in Fig. 2c, can the authors suggest that "the interaction with RIPK3 is not sufficient for FRET induction?"

RESPONSE: To clarify this point, we now changed the sentences to, "**Further refinement of α 14 to generate SMART**

Although α 14 monitored necroptosis, transient transfection of α 14 blocked cell proliferation (data not shown). To investigate the mechanisms how α 14 monitors necroptosis and circumvent this drawback, we generated a further series of α 14 mutants. To test whether binding of α 14 to RIPK3 is required for monitoring necroptosis, we first mutated the phenylalanine at position 234 of MLKL, that is critical for RIPK3 binding in vitro, to glutamic acid (F234E) (Fig. 2a). Unexpectedly, an F234E mutant of α 14 still interacted with RIPK3 and showed the increase in the FRET/CFP ratio upon TZ stimulation (Fig. 2b, c). We next replaced amino acids at the indicated fragments with a set of four flexible Ser-Ala-Gly-Gly (SAGG) repeats to maintain the same spacing between Ypet and CFP (Fig. 2a, Supplementary Fig. 2). The TZ-induced increase in the FRET/CFP ratio was partially or completely abolished in cells expressing $\Delta\alpha$ 2 α 3, $\Delta\alpha$ 3, or Δ abc. In sharp contrast, TZ increased the FRET/CFP ratio of Δ a and Δ ab comparable to that of α 14 (Fig. 2b, Supplementary Fig. 3a, b). Notably, $\Delta\alpha$ 2 α 3 did not interact with RIPK3 or show the increase in the FRET/CFP ratio (Fig. 2b, c). Three biosensors including α 19, α 59, and Δ abc bound to RIPK3, but did not show the increase in the FRET/CFP ratio (Fig. 2c), suggesting that the interaction of RIPK3 with the FRET biosensors is prerequisite, but not sufficient for monitoring necroptosis. Among FRET biosensors showing the increase in the FRET/CFP ratio upon TZ stimulation, we were only able to obtain cells stably expressing Δ ab. Thus, this construct is henceforth referred to as SMART (a Sensor for MLKL activation by RIPK3 based on FRET). (Lines 160-179).

COMMENT: Line 176-181: I understand that the kinase activity of RIP3 is not necessary for the interaction with SMART with RIPK3, but is required for the SMART activation. Please rewrite the paragraph from Line 159 more concisely to clarify the point.

RESPONSE: As suggest, we changed the sentences to " Cotransfection experiments revealed that SMART and a mutant of SMART where all putatively phosphorylated serines and threonine were replaced with alanine (4ST5A), still

interacted with RIPK3 in the absence or presence of GSK'872 (Fig. 2d). Thus, neither the kinase activity of RIPK3 nor phosphorylation of SMART is required for their binding." (Lines 180-183).

COMMENT: Line 427: I believe that Histone H3 is released from apoptotic cells as a component of nucleosomes in apoptotic bodies. The comparison here may not be appropriate.

RESPONSE: As suggested, we deleted the sentence.

Reviewer #2 (Remarks to the Author):

The authors have addressed most of my points.

The following points still need to be addressed, in the light of introduction of a new model for SMART activation:

COMMENT: 1) Since the model of SMART activation has changed from being dependent on RIPK3-MLKL binding to MLKL oligomerization, this should be reflected in the wording in the abstract. The statement "Activation of SMART requires endogenous RIPK3 and MLKL, and depends on oligomers of MLKL." is not correct, since Q343A mutant can also activate the SMART probe in the absence of RIPK3. In other words, mechanistically, SMART activation depends on oligomerization of MLKL, which is induced by RIPK3 or other means, such as mutational activation.

RESPONSE: As suggested, we changed the sentence to "SMART monitors plasma membrane translocation of MLKL, which is induced by RIPK3 or mutational activation." (Lines 45-47).

COMMENT: 2) The phrase "and inhibits necroptosis through preventing oligomers of MLKL in human cells" (line 248) is not correct. NSA inhibits MLKL function by preventing its plasma membrane localization. Also, the ref 25 at the end of that sentence is not correctly placed. NSA was not

shown to inhibit MLKL oligomerization in that study. Which study shows that NSA inhibits MLKL oligomerization?

RESPONSE: We appreciate pointing out our misunderstanding the mechanisms underlying NSA-dependent suppression of necroptosis. Now we changed the sentence to “NSA targets cysteine at 86 (C86) of the N-terminal domain (NTD) of human, but not murine MLKL, and inhibits necroptosis through preventing plasma membrane translocation of oligomerized MLKL.” (Lines 254-256).

3) The authors should provide evidence that NSA inhibited oligomerization of MLKL in the presence of the hSMART probe to prove that hSMART works by detecting MLKL oligomerization. A simple way to do this is to employ the non-reducing PAGE system used by Wang et al., 2014, Mol Cell study. NSA does not seem to block oligomerization of MLKL in that study.

RESPONSE: As suggested, we stimulated HT29-SMART cells with TBZ in the absence or presence of NSA and cell lysates were subjected to SDS-PAGE under non-reducing conditions. We found that NSA did not block oligomerization of MLKL (see Rebuttal Figure 1), but blocked necroptosis and suppressed the increase in the FRET/CFP ratio of SMART. Thus, we concluded that MLKL does not monitor oligomerization of MLKL itself. Please also see the response to the comment 6) below. We only included the Rebuttal Figure 1 in a rebuttal letter.

Rebuttal Figure 1. NSA does not block oligomerization of MLKL. HT29-SMART cells were stimulated with TBZ in the absence or presence of NSA for the indicated times. Cell lysates were subjected to non-reducing SDS-PAGE (a) or reducing SDS-PAGE (b). Expression of MLKL and SMART (GFP) was determined by immunoblotting with anti-MLKL and GFP antibodies, respectively. Oligo and mono indicate oligomers and monomer of MLKL, respectively. Notably, anti-GFP antibody detected SMART on SDS-PAGE under reducing (b), but not non-reducing conditions (a).

4) NSA blocking SMART activation suggests that MLKL plasma membrane translocation is required for the activation of the sensor.

RESPONSE: Please see the response to the comment 6).

5) Does T357E/S358D-induced MLKL oligomerization and hSMART activation get blocked by NSA?

RESPONSE: We recently showed that this mutant does not kill human cells and is actually incapable of killing cells (Figure 5c-f, Petrie et al, Nat Commun 2018).

6) The key experiment to support the new model is to do non-reducing PAGE after TSZ ± NSA and immunoblot for the SMART probe (e.g. CFP), showing that the probe is associated with the endogenous MLKL oligomers and that NSA is blocking this association. Alternatively, a size-exclusion chromatography experiment will be needed to support the model. Currently, the only evidence that supports the new model is the Q343A experiment and this experiment lacks evidence that the oligomerization, but not the plasma membrane translocation of MLKL is required for SMART activation. The HT-29 ± NSA experiment does not prove the oligomerization model, since NSA blocks plasma membrane translocation.

RESPONSE: As suggested, we performed non-reducing SDS-PAGE and the membrane was immunoblotted with anti-MLKL and then reblotted with anti-GFP antibodies. While anti-GFP antibody efficiently detected SMART under reducing conditions (Rebuttal Figure 1b), at least in our hand, anti-GFP antibody could not detect SMART under non-reducing conditions (Rebuttal Figure 1a). Thus, we could not determine whether SMART was incorporated into the oligomerized MLKL under these experimental conditions.

Rebuttal Figure 1. NSA does not block oligomerization of MLKL. HT29-SMART cells were stimulated with TBZ in the absence or presence of NSA for the indicated times. Cell lysates were subjected to non-reducing SDS-PAGE (**a**) or reducing SDS-PAGE (**b**). Expression of MLKL and SMART (GFP) was determined by immunoblotting with anti-MLKL and GFP antibodies, respectively. Oligo and mono indicate oligomers and monomer of MLKL, respectively. Notably, anti-GFP antibody detected SMART on SDS-PAGE under reducing (**b**), but not non-reducing conditions (**a**).

To further investigate the mechanisms how SMART monitor necroptosis, we treated HT29-SMART cells with TBZ in the absence or presence of NSA, and cytosolic and membrane fractions were subjected to Blue-Native PAGE. As expected, MLKL formed oligomers on the plasma membrane of cells at 6 hours following TBZ stimulation (Supplementary Figure 7a). In sharp contrast, NSA blocked plasma membrane translocation of oligomerized MLKL and an increase in the FRET/CFP ratio of SMART. Thus, SMART monitors

plasma membrane translocation of oligomerized MLKL, but not oligomerized MLKL itself. Since anti-GFP antibody did not work well on BN-PAGE, we could not detect SMART by anti-GFP antibody in the cytoplasmic fraction on SDS-PAGE under reducing conditions (Figure 7b). This suggests that SMART did not translocate into the plasma membrane via direct interaction with oligomerized MLKL. This result was also consistent with that FRET signals of SMART were diffusely distributed in the cytosol of cells undergoing necroptosis (Figure 3a). One of the plausible explanations would be that SMART might monitor drastic changes of cellular conditions induced by plasma membrane translocation of oligomerized MLKL, such as relative ratios of RIPK3 and MLKL in the cytosol. Further study will be required to address this issue. We made a new Supplementary Figure 7 to include the results and mentioned it in the text (Lines 249-258 and 389-396).

MuraiFigS7

Supplementary Figure 7. NSA blocks plasma membrane translocation of oligomerized MLKL. HT29-SMART cells were stimulated with TBZ in the absence or presence of NSA for the indicated times. Then cytosolic and membrane fractions were prepared and subjected to BN-PAGE (**a**) or SDS-PAGE under reducing conditions (**b**). Expression of each protein in

cytosolic (C) or membrane (M) fractions was determined by immunoblotting with the indicated antibodies. Oligo and mono indicate oligomers and monomer of MLKL, respectively. Results are representative of two independent experiments.

REVIEWERS' COMMENTS:

Reviewer #2 (Remarks to the Author):

I am satisfied with the revision.